# Landscape of histone modifications in a sponge reveals the origin of animal *cis*-regulatory complexity

**Federico Gaiti**[†‡], **Katia Jindrich**, **Selene L Fernandez-Valverde**[§], **Kathrein E Roper**, **Bernard M Degnan***, **Miloš Tanurdžić***

School of Biological Sciences, University of Queensland, Brisbane, Australia

*For correspondence: b.degnan@uq.edu.au (BMD); m.tanurdzic@uq.edu.au (MT)

Present address: [†]New York Genome Center, New York City, United States; [‡]Department of Medicine, Weill Cornell Medicine, New York City, United States; [§]Consejo Nacional de Ciencia y Tecnología, Laboratorio Nacional de Genómica para la Biodiversidad, Centro de Investigación y de Estudios Avanzados del IPN, Guanajuato, México

Competing interests: The authors declare that no competing interests exist.

**Abstract** Combinatorial patterns of histone modifications regulate developmental and cell type-specific gene expression and underpin animal complexity, but it is unclear *when* this regulatory system evolved. By analysing histone modifications in a morphologically-simple, early branching animal, the sponge *Amphimedonqueenslandica*, we show that the regulatory landscape used by complex bilaterians was already in place at the dawn of animal multicellularity. This includes distal enhancers, repressive chromatin and transcriptional units marked by H3K4me3 that vary with levels of developmental regulation. Strikingly, *Amphimedon* enhancers are enriched in metazoan-specific microsyntenic units, suggesting that their genomic location is extremely ancient and likely to place constraints on the evolution of surrounding genes. These results suggest that the regulatory foundation for spatiotemporal gene expression evolved prior to the divergence of sponges and eumetazoans, and was necessary for the evolution of animal multicellularity.

## Introduction

Animals rely on genomic regulatory systems to direct the dynamic spatiotemporal and cell type-specific gene expression that is essential for the development and maintenance of a multicellular lifestyle. However, *how* such a system originated and evolved in animals remains unclear. As the last common ancestor of modern animals already possessed an extensive repertoire of regulatory genes, including most transcription factors and signaling pathways used in bilaterian development (*Srivastava et al., 2010*; *Larroux et al., 2008*; *Degnan et al., 2009*; *Larroux et al., 2006*; *Richards and Degnan, 2009*; *Ryan et al., 2013*; *Moroz et al., 2014*; *King et al., 2008*; *Sebé-Pedrós et al., 2011*; *de Mendoza et al., 2013*; *King et al., 2003*; *Richter and King, 2013*), the evolution of animal multicellularity likely required more than the origin of novel genes. Other regulatory features, such as *cis*-regulatory DNA and combinatorial patterns of histone covalent post-translational modifications (PTMs) (*Davidson and Peter, 2015*), would have been instrumental to direct differential gene expression in the first multicellular animals. For instance, recent analysis of the genome of *Capsaspora*, one of the closest unicellular relatives of animals, reveals a lack of chromatin repressive marks, developmental promoter types and distal *cis*-regulatory elements (enhancers) typically present in complex animals (*i.e.*, eumetazoans) (*Sebé-Pedrós et al., 2016*).

The development of high-throughput chromatin assays like chromatin immunoprecipitation coupled with massively parallel sequencing (ChIP-seq) (*Robertson et al., 2007*) has allowed the dissection of chromatin-encoded information beyond the primary DNA sequence, especially the systematic examination of histone PTMs and their role(s) in transcriptional regulation (*Zhou et al., 2011*; *Thurman et al., 2012*; *Kundaje et al., 2015*; *ENCODE Project Consortium, 2012*). Although combinatorial patterns of histone acetylation and methylation are key components of gene

**eLife digest** Animals come in many shapes and sizes, and vary in how they move, grow and reproduce. The long-held thought that animal complexity is related to the number of genes that are in the animal's DNA has now been largely dismissed; simple animals like sponges and cnidarians (for example, jellyfish, anemones and corals) have comparable gene numbers to vertebrates, insects, mollusks and other complicated bilaterians (animals that feature a plane of symmetry, meaning that they have a top, a bottom, a front and a back). This observation led to the idea that gene regulation (how and when genes are turned off and on) is responsible for the evolution of animal diversity.

Genomic DNA packs into cells by winding around proteins called histones. Histones themselves can bear certain chemical marks, which in turn determine if the genes contained in the DNA associated with the histones are going to be turned on or off. In bilaterians and cnidarians these marks substantially contribute to gene regulation. Some of these marks predate the evolution of multicellular animals from single-celled organisms. However, the origin of the marks that associate with the gene regulatory elements that are essential for animals to be multicellular remained unknown. In other words, does the evolution of histone marks underpin animal complexity?

Gaiti et al. turned to the marine sponge *Amphimedon queenslandica* to address this question. Sponges are one of the morphologically simplest animals, lacking a gut, nerves and muscles. By analyzing histone marks in this sponge, Gaiti et al. found they were remarkably similar to the networks of histone marks seen in more complex animals. This is consistent with this form of gene regulation being present at the dawn of the animal kingdom. Indeed, this mode of gene regulation may have been necessary for multicellular animals to first evolve.

It now appears that most of the genes and regulatory mechanisms underlying the formation of complex animals, like ourselves, had an unexpected early origin – probably as early as the first steps in the evolution of multicellular animals from single-celled organisms. Further studies of animals that are close relatives of sponges, such as comb jellies, and their single-celled cousins, may further improve our understanding of how these simple single-celled organisms became multicellular animals.

regulatory mechanisms underpinning the formation and maintenance of eumetazoans (*Schwaiger et al., 2014*), it remains unknown if this system is restricted to these animals or is indeed more ancient.

Porifera (sponges) are considered one of the oldest surviving phyletic lineages of animals, diverging from other metazoans around 700 Mya (*Erwin et al., 2011*). Despite being one of the morphologically simplest animals, lacking a gut, nerves and muscles, sponges possess an extensive gene repertoire for transcriptional regulation required in eumetazoan development and body patterning (*Srivastava et al., 2010*; *Larroux et al., 2008*, *2006*; *Adamska et al., 2007*; *Gaiti et al., 2015*; *Nakanishi et al., 2014*; *Conaco et al., 2012*; *Riesgo et al., 2014*; *Grimson et al., 2008*; *Richards et al., 2008*; *Leininger et al., 2014*; *Fortunato et al., 2015*, *2014*; *Bråte et al., 18212015*). Here, following on from our recent transcriptomic studies that revealed that the sponge *Amphimedon queenslandica* (herein *Amphimedon*) has dynamic developmental gene expression akin to eumetazoans (*Gaiti et al., 2015*; *Fernandez-Valverde et al., 2015*; *Levin et al., 2016*), we set out to determine whether this transcriptional complexity is paralleled by regulatory complexity encoded by combinatorial histone PTM patterns. By analysing an extensive ChIP-seq compendium of histone H3 PTMs in this sponge, we show that a complex gene regulatory landscape comprised of combinatorial histone modifications was already in place at the dawn of animals. Moreover, we provide evidence for the evolution and expansion of distal *cis*-regulatory genomic capabilities at the origin of the animal kingdom.

## Results

### *Amphimedon* key regulatory chromatin states are shared with eumetazoans

We carried out chromatin immunoprecipitation (ChIP) on sexually reproducing *Amphimedon* adults and larvae using antibodies against specific histone H3 PTMs that have been used to define chromatin states in model bilaterians (*Zhou et al., 2011*; *Ho et al., 2014*) (*Figure 1A*). These analyses were undertaken on separate admixtures of adult and larval somatic cell types and, thus, a diversity of gene transcriptional states. Importantly, *Amphimedon* adults and larvae are comprised of different cell types with markedly different transcriptional profiles and regulatory states (*Gaiti et al., 2015*; *Conaco et al., 2012*; *Fernandez-Valverde et al., 2015*; *Degnan et al., 2015*). While our sampling strategy increases the biological complexity of chromatin states *in toto*, it may dilute cell type-specific signals. This contrasts with ChIP-seq analyses performed on cell lines, embryos with few cell types, or distinct tissue samples, which encapsulate more homogenous cellular populations and environments (*Sebé-Pedrós et al., 2016*; *Kundaje et al., 2015*; *Schwaiger et al., 2014*; *Gerstein et al., 2010*; *Pérez-Lluch et al., 2015a*). Given the current *Amphimedon* genome is a draft sequence, our analyses may also be incomplete in regions that have incomplete annotations and gaps in the assembly (13% of the total genome assembly) (*Srivastava et al., 2010*).

The antibodies used target the following histone H3 PTMs: (i) monomethylated lysine 4 (H3K4me1), associated with distal *cis*-regulatory elements such as enhancers; (ii) trimethylated lysine 4 (H3K4me3), enriched in active promoters; (iii) trimethylated lysine 36 (H3K36me3), found with actively transcribed regions; (iv) trimethylated lysine 27 (H3K27me3), enriched in Polycomb-silenced regions; and (v) acetylated lysine 27 (H3K27ac), which occurs around activated regulatory regions. We also used an antibody against total histone H3 (*Figure 1—source data 1*). An antibody against unphosphorylated Ser2 residues of RNA polymerase II (RNAPII 8WG16) C-terminal domain also was included (*Brookes and Pombo, 2009*) (*Figure 1—source data 1*). As the entire amino acid sequence of histone H3 is perfectly conserved in *Amphimedon*, along with the relevant histone methyltransferases and acetyltransferases, these antibodies are predicted to recognize the correct epitopes (*Figure 1—figure supplement 1*; *Figure 1—source data 2*; *Figure 1—source data 3*). These antibodies recognize the correct epitopes in even more distantly related organisms (*i.e.*, non-metazoan eukaryotes) (*e.g.*, [*Sebé-Pedrós et al., 2016*; *Ercan et al., 2009*; *Barraza et al., 2015*; *Harmeyer et al., 2015*; *Liu et al., 2007*; *Eckalbar et al., 2016*]).

ChIP-seq reads generated from immunoprecipitated and input (whole-cell extract) DNA were aligned to the *Amphimedon* genome (*Srivastava et al., 2010*), resulting in highly reproducible data sets (*Figure 1—figure supplement 2*; *Figure 1—source data 4*; *Figure 1—source data 5*). Uniquely mapped reads were subsequently used to identify a set of distinct chromatin states based on the five different histone H3 PTMs we assayed. Specifically, chromatin states were predicted throughout the genome training a multivariate Hidden Markov Model with different *a priori* defined states (from 5 to 15) (Materials and methods). We elected to use a 9-state model for all further analyses as it covered all major gene coding and regulatory components (promoter, enhancer, gene body) that we expected to resolve with this selection of histone H3 PTMs. Despite the inherent cellular heterogeneity of our starting material, we were able to resolve specificities towards gene components between these nine chromatin states. They fell into two broad categories: one that correlated with actively transcribed genes that include active promoters ('TssA') and enhancers ('TxEnhA', 'EnhWk'), and 5' and 3' boundaries of transcribed genes ('TxFlnk'); and another category with genes with no or little detectable transcription; these include bivalent or poised regulatory ('BivTx', 'EnhP'), repressed Polycomb ('ReprPC', 'ReprPCWk'), and quiescent ('Quies') states (*Figure 1B–D*). The nine chromatin states differentially associated with specific *Amphimedon* genomic features. For instance, the 'TssA' state (defined by the presence of H3K4me3) was enriched around transcription start sites (TSSs) of active genes. 'TxEnhA' state (defined by H3K4me1, H3K27ac, and H3K36me3 enrichment) associated with coding exons and introns that correspond to potential *cis*-regulatory elements and short intergenic regions, which are common in the *Amphimedon* genome (*Kundaje et al., 2015*; *Fernandez-Valverde et al., 2015*; *Kowalczyk et al., 2012*; *Ritter et al., 2012*; *Singer et al., 2015*; *Birnbaum et al., 2012*; *Zentner and Scacheri, 2012*; *Zentner et al., 2011*; *Fernandez-Valverde and Degnan, 2016*). In contrast, the 'ReprPC' states (defined by H3K27me3 enrichment) were spread through the gene bodies of repressed genes, consistent with the known role of

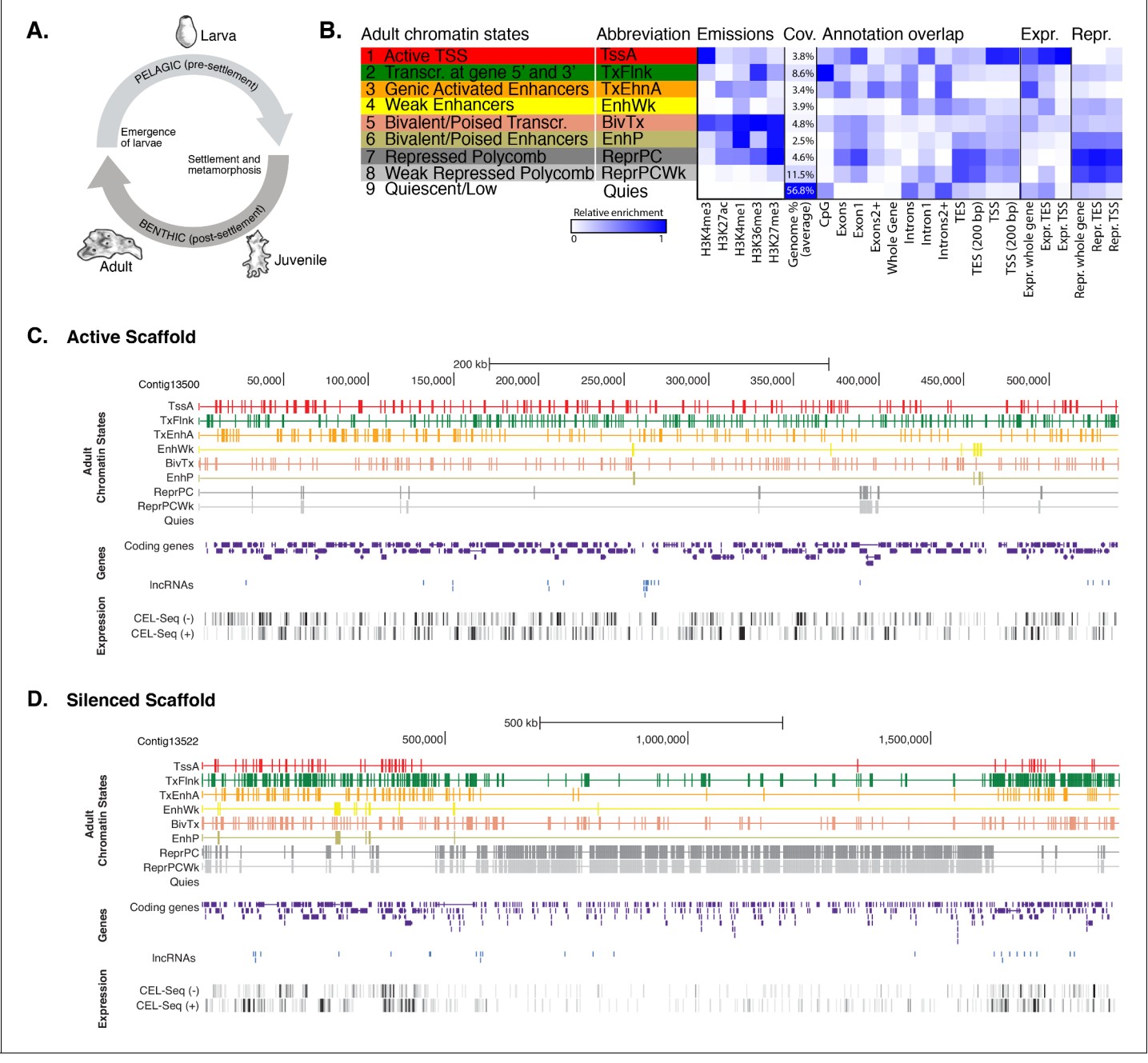

**Figure 1.** Chromatin states in Amphimedon. (A) Schematic representation of *Amphimedon* life cycle. Larvae (oval shaped, 300–500 μm long) emerge from maternal brood chambers and then swim in the water column before they develop competence to settle and initiate metamorphosis into a juvenile. The juvenile body plan, which displays the hallmarks of the adult body plan, including an aquiferous system with canals, choanocytes chambers and oscula, is the outcome of the dramatic reorganization of the radially-symmetrical, bi- or trilayered larva. This juvenile will then grow and mature into a benthic adult (ranging from 10–30 cm³) (***Degnan et al., 2015***; ***Edgar et al., 2002***). (B) Definition and enrichments for a 9-state Hidden Markov Model based on five histone PTMs (H3K4me3, H3K27ac, H3K4me1, H3K36me3 and H3K27me3) in adult *Amphimedon*. From left to right: chromatin state definitions, abbreviations, histone PTM probabilities, genomic coverage, protein-coding gene functional annotation enrichments, expressed (Expr.) and repressed (Repr.) protein-coding gene enrichments. Blue shading indicates intensity, scaled by column. (C) Adult chromatin state annotations on gene rich highly transcribed (active) scaffold (contig13500) showing the predominance of 'TssA', 'TxFlnk', and 'TxEnhA' states. For the definition of chromatin states see panel (A). Coding genes (purple) and long non-coding RNAs (blue) are shown, along with signal coverage tracks showing CEL-seq expression in adult. A grey scale indicates CEL-seq expression level: white (no-expression); black (highest expression). (D) Adult chromatin state annotations on a predominantly silenced scaffold (contig13522 from 500,000 to 1,500,000 bp) showing the prevalence of 'ReprPC' and 'ReprPCWk'

*Figure 1 continued on next page*

*Figure 1 continued*

states. For the definition of chromatin states see panel (**A**). Coding genes (purple) and long non-coding RNAs (blue) are shown, along with signal coverage tracks showing CEL-seq expression in adult. A grey scale indicates CEL-seq expression level: white (no-expression); black (highest expression).

The following source data and figure supplements are available for figure 1:

**Source data 1.** Histone H3 covalent post-translation modifications and RNA Polymerase II investigated in this study and their typical genomic localization relative to coding genes and regulatory regions in bilaterian model organisms.

**Source data 2.** Histone H3 sequences used to generate *Figure 1—figure supplement 1*.

**Source data 3.** BLASTp search outcome of the relevant histone methyltransferases and acetyltransferases against *Amphimedon queenslandica* proteins (NCBI nr database; *E*-value <1e-09).

**Source data 4.** Summary statistics and quality metrics of the ChIP-seq datasets used in this study.

**Source data 5.** Validation of the ChIP-seq results by ChIP-quantitative PCRs (ChIP-qPCRs).

**Figure supplement 1.** Multiple sequence alignment of various eukaryotic histone H3 proteins (1–136 amino acids), produced by using ClustalO (RRID: SCR_001591) (*Sievers et al., 2011*).

**Figure supplement 2.** Assessment of reproducibility for biological replicates between histone modifications andRNA Polymerase II.

**Figure supplement 3.** Neighborhood positional enrichment plots of adult chromatin states around transcription start site (TSS) and transcription end site (TES) of proteins-coding genes, produced by ChromHMM (*Ernst and Kellis, 2012*).

**Figure supplement 4.** Chromatin states in *Amphimedonlarva*.

---

H3K27me3 in transcriptional silencing (*Zhou et al., 2011*; *Ho et al., 2014*) (*Figure 1BD*; *Figure 1— figure supplement 2*; *Figure 1—figure supplement 3*).

Despite being comprised of different cell types and having a distinct gene expression profile from the adult, the larval genome possesses a remarkably similar set of chromatin states (*Figure 1— figure supplement 4*). Obtaining consistent chromatin states based on histone PTMs ChIP-seq data from two markedly different stages of the *Amphimedon* life cycle provides corroborating evidence that this sponge possesses the same regulatory states as present in eumetazoans.

## Histone PTMs and the tuning of gene expression in *Amphimedon*

To investigate the distribution of histone H3 PTMs in *Amphimedon* genes, we calculated the average enrichment of histone H3 PTMs and RNAPII relative to the TSSs of protein-coding genes. Input-normalized ChIP-seq read coverage revealed a strong unimodal H3K4me3 peak positioned immediately after the TSS of expressed genes that co-localizes with H3K27ac and RNAPII (*Figure 2A*; *Figure 2— figure supplement 1*; *Figure 2—figure supplement 2A*). Additionally, H3K4me3 marked (i) genes with head-to-head orientation that may be under the control of a bidirectional promoter (a common feature in the *Amphimedon* genome [*Fernandez-Valverde and Degnan, 2016*]), and (ii) alternative TSSs (*Figure 2—figure supplement 3*). This is consistent with H3K4me3 being promoter-proximal and positioned on the +1 nucleosome (*Zhou et al., 2011*; *Ho et al., 2014*; *Lenhard et al., 2012*). A prominent nucleosome-depleted region was observed right upstream of the TSS of expressed genes (likely corresponding to the proximal promoter) followed by a narrowly localized nucleosome (the +1 nucleosome) (see below *Figure 2—figure supplement 4D*), suggesting that the interplay between nucleosome positioning and transcription is conserved in sponge promoters (*Sebé-Pedrós et al., 2016*; *Schwaiger et al., 2014*; *Roy et al., 2010*; *Bai and Morozov, 2010*; *Jiang and Pugh, 2009*). Overall, the distribution of histone H3 PTMs in *Amphimedon* correlated with the expression state of its genes, as in eumetazoans (*Schwaiger et al., 2014*; *Roy et al., 2010*) (Fisher's exact test, FDR adjusted *p*-value<0.05) (*Figure 2B and C*; *Figure 2—figure supplement 2B–D*).

To investigate the dynamics of histone PTMs in genes regulated throughout *Amphimedon* development, we analysed CEL-seq data (*Levin et al., 2016*; *Hashimshony et al., 2012*; *Anavy et al.,*

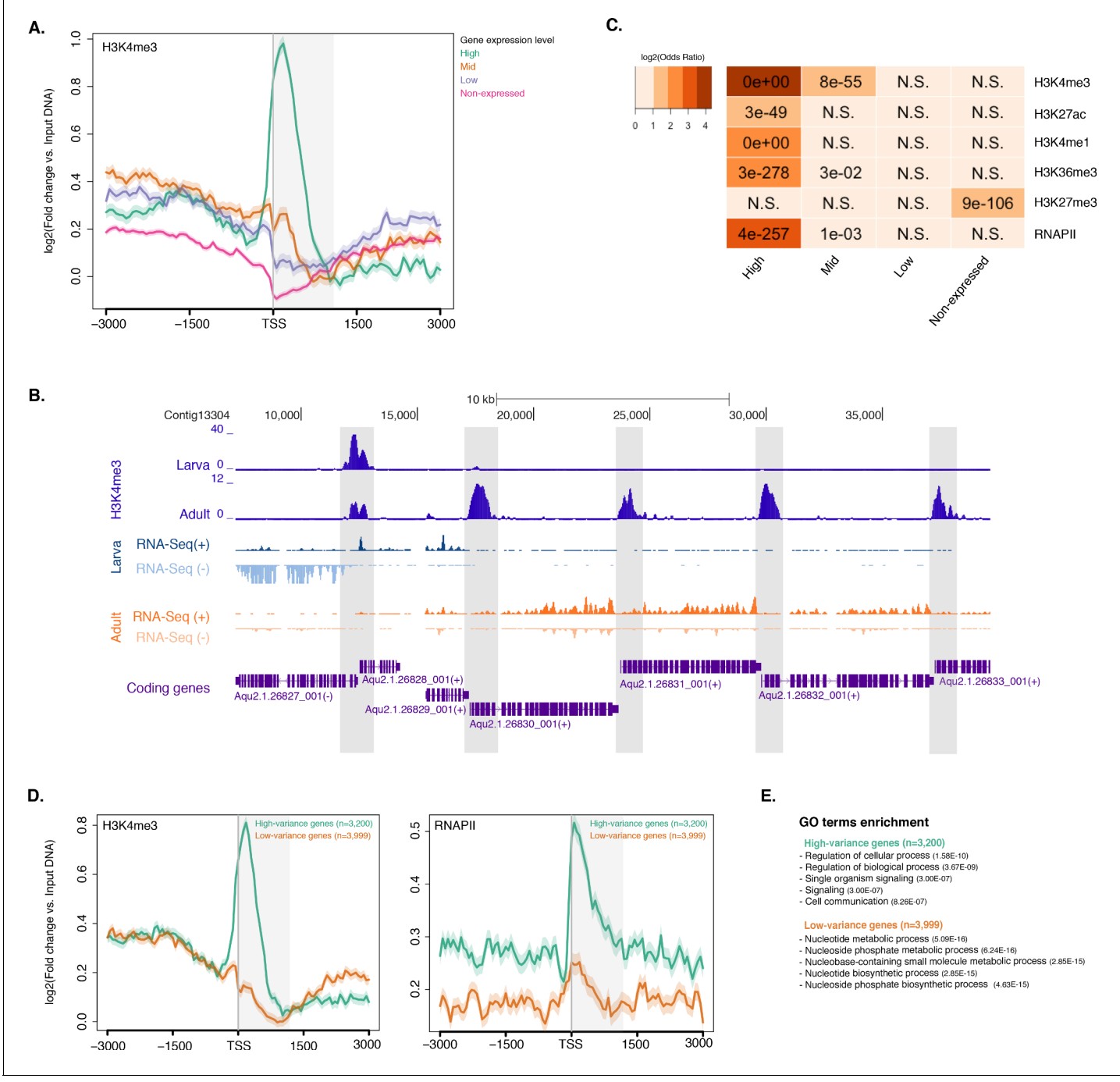

**Figure 2.** Histone PTMs are correlated with gene expression variations during development. (A) TSS-centred average input DNA normalised read coverage plot of H3K4me3 across *Amphimedon* protein-coding genes. The x-axis spans ± 3 kb around TSSs and represents the position within the gene relative to TSS. The y-axis represents the input DNA normalised enrichment for H3K4me3 ChIP-seq reads in adult *Amphimedon*. Pink line: Non-expressed genes. Blue line: Low expressed genes. Orange line: Medium expressed genes. Light blue line: High expressed genes. The shaded gray area represents the average size of *Amphimedon* coding sequences. (B) Example of coding genes marked by H3K4me3 peaks. The genomic window shows input DNA-normalized H3K4me3 coverage and RNA-seq expression in both larva and adult. (C) The association of regions of enrichment of five histone H3 PTMs (H3K4me3, H3K27ac, H3K4me1, H3K36me3 and H3K27me3) and RNAPII with lists of various gene expression groups in adult is shown. The color key represents the log2(odds ratio) and the significant adjusted *P*-values (Fisher's exact test) are superimposed on the grids. A *P*-value of zero means the overlap is highly significant. N.S.: not significant. Odds ratio represents the strength of association. (D) TSS-centred average input DNA normalised read coverage plots of H3K4me3 and RNAPII across 'high-variance' and 'low-variance' protein-coding genes. The x-axis spans ± 3 kb around TSSs and represents the position within the gene relative to TSS. The y-axis represents the input DNA normalised enrichment for ChIP-seq

*Figure 2 continued on next page*

*Figure 2 continued*

reads in adult *Amphimedon*. Light blue: high-variance coding genes. Orange line: low-variance coding genes. The shaded gray area represents the average size of *Amphimedon* coding sequences. (**E**) Top five most significantly enriched Gene Ontology (GO) terms for high-variance and low-variance protein-coding genes (adjusted *P*-values in brackets, Hypergeometric test). The full GO table is shown in *Figure 2—source data 1*.

The following source data and figure supplements are available for figure 2:

**Source data 1.** GO biological process term enrichment outcome for the high-variance and low-variance gene sets (Hypergeometric test, FDR<0.01).
**Source data 2.** KEGG pathways significantly enriched in low-variance and high-variance genes.
**Figure supplement 1.** TSS-centred average input DNA normalised read coverage plots and heatmaps of RNAPII, H3K27ac, H3K36me3, H3K4me1 and H3K27me3 across *Amphimedon* protein-coding genes.
**Figure supplement 2.** Histone PTMs and gene expression variations during development.
**Figure supplement 3.** H3K4me3 enrichment at genes with head-to-head orientation and alternative TSSs.
**Figure supplement 4.** ChIP-seq profiles of H3K4me3 and total histone H3 across high-variance and low-variance genes.

*2014*), comprising of 82 *Amphimedon* developmental samples from early cleavage to adult compressed into 17 stages, in the context of ChIP-seq profiles of total histone H3, H3K4me3, and RNA-PII. We selected genes with the highest median absolute deviation for gene expression across these 17 *Amphimedon* developmental stages (effectively measuring the amplitude of change in expression levels for a given gene), resulting in a set of 3,200 'high-variance' expressed genes (*Figure 2—figure supplement 4A*). The remaining expressed genes were defined as 'low-variance' genes (3,999) (see Materials and methods for the complete list of selection criteria). It is noteworthy that the high-variance genes were, on average, also expressed at higher levels than the low-variance genes (average adult expression of 51 vs 7 CEL-seq normalized counts, respectively). The TSSs of high-variance genes were strongly marked by H3K4me3 and occupied by RNAPII (*Figure 2D*; *Figure 2—figure supplement 4B*). Additionally, they showed nucleosome depletion right upstream of the TSSs (seen as lack of total histone H3 signal), consistent with the notion that H3K4me3 near TSSs destabilizes the interaction between histones and DNA to direct RNAPII to facilitate binding of promoter regulator elements and initiate transcription (*Jiang and Pugh, 2009*; *Ha et al., 2011*; *Boeger et al., 2003*) (*Figure 2—figure supplement 4D*). Conversely, lower levels of H3K4me3 or RNAPII (Mann-Whitney U test, p-value=0.05287 and p-value<2.2e-16, respectively; *Figure 2D*; *Figure 2—figure supplement 4C*) but higher nucleosome occupancy characterized low-variance genes (seen as lack of nucleosome depletion right upstream of the TSSs; *Figure 2—figure supplement 4D*). These results are consistent with H3K4me3 being predictive of gene expression levels (*Ha et al., 2011*; *Karlić et al., 2010*).

The distinctive landscapes of histone PTMs in high-variance and low-variance genes also correlated with distinct functional related gene groups, as indicated by Gene Ontology (GO)and KEGG pathway analyses. High-variance genes, which also include a significantly higher number of transcription factor gene families (*e.g.*, JUN and ATF6 *Jindrich and Degnan [2016]*) compared to low-variance genes (Fisher's exact test, p-value=3.872e-08), were predominantly enriched in signaling pathways (Hypergeometric test, FDR adjusted p-value<0.01; *Figure 2E*; see *Figure 2—source data 1* and *Figure 2—source data 2* for the complete list). In contrast, low-variance genes were enriched for metabolic GO terms (*Figure 2E*; see *Figure 2—source data 1* and *Figure 2—source data 2* for the complete list). This result is consistent with H3K4me3 being important for tuning the gene expression of dynamically expressed developmental genes, *e.g.*, transcription factor and signaling genes. However, it remains unclear whether H3K4me3 is needed for high levels of gene expression or if it is needed for, or associated with, frequent switching of transcriptional status.

## Absence of H3K4me3 in strongly developmentally regulated genes appear to be a metazoan conserved feature

The recent finding that transcription of a subpopulation of extremely dynamically expressed genes – typically being expressed at only one stage of development – in *Drosophila* and *C. elegans* occurs in the absence of H3K4me3 challenged the canonical role of histone PTMs in the modulation of gene expression (*Pérez-Lluch et al., 2015a*). To test whether this newly-discovered feature is conserved in non-bilaterians, we interrogated above-mentioned CEL-seq data (*Levin et al., 2016*; *Hashimshony et al., 2012*; *Anavy et al., 2014*), comprising 82 *Amphimedon* developmental samples from early cleavage to adult compressed into 17 stages, and arbitrarily selected, similarly to *Pérez-Lluch et al. (2015a)*, the 1,000 genes with the lowest coefficients of variation ('*stable*' genes) expressed with minor changes throughout development. Conversely, the 1,000 genes with the highest coefficients of variation were defined as '*regulated*' genes. Notably, the '*regulated*' genes consisted of a small population of genes that differed from the '*high-variance*' genes described earlier in having much more restricted expression patterns, mainly expressed at late juvenile and/or adult stage (*Figure 3—figure supplement 1*). Although stable and regulated genes had similar levels of RNAPII and total histone H3 (*Figure 3—figure supplement 1B and C*), the stable genes were strongly marked by H3K4me3 and the regulated genes had significantly lower levels of H3K4me3 (Mann-Whitney U test, p-value=7.431e-05; *Figure 3A*), suggesting that reduction in H3K4me3 levels does not affect expression of the regulated genes (*Pérez-Lluch et al., 2015a*).

We compared the pattern of H3K4me3 between one of the top three stably expressed genes during sponge development (Aqu2.1.40735_001, a E3 ubiquitin-protein ligase), and the gene with the highest coefficient of variation (Aqu2.1.39666_001, a putative sponge-specific gene specifically expressed in adult) (*Figure 3B*). The former showed a strong H3K4me3 enrichment at the TSS, whereas the latter lacked any marking, though its expression in the adult was ~70 times higher than the stable gene (33 vs 2361 CEL-seq normalized counts in adult, respectively). This lack of H3K4me3 at the TSS of regulated genes was similarly observed in the larva, exemplified here by a larva-specific regulated gene (Aqu2.1.34366_001) expressed 3.5-fold higher than the above-mentioned stable gene (Aqu2.1.40735_001) (147 vs 43 CEL-seq normalized counts in larva, respectively) (*Figure 3B*). Additionally, as shown in *Drosophila* (*Pérez-Lluch et al., 2015a*), regulated genes showed higher levels of H3K27me3 (Mann-Whitney U test, p-value<6.517e-06) and lower levels of H3K36me3 (Mann-Whitney U test, p-value<9.235e-08) than did stable genes (Figure 3—figure supplement 1D and E). Analyzing RNA-seq–based gene expression through the development of the cnidarian *Nematostella vectensis* (*Helm et al., 2013*) and previously published ChIP-seq data sets in *Nematostella* adult female polyps (*Schwaiger et al., 2014*), we obtained the same pattern (Mann-Whitney U test, p-value<2.2e-16; *Figure 3C*).

These results suggest that H3K4me3 might not be instrumental for extremely dynamic developmental expression and enforces our interpretation that it is required for tuning the levels of gene expression, a pattern that appears to be a conserved metazoan feature (*Pérez-Lluch et al., 2015a*).

## Polycomb repressive complex 2 (PRC2) is conserved in *Amphimedon* and its binding sites contain putative GAGA factor binding motifs

PRC2 is responsible for the trimethylation of lysine 27 of histone H3 (H3K27me3), one of the best-characterized repressive histone H3 PTMs (*Margueron and Reinberg, 2011*). As a step to investigate a putative mechanism of PRC2-mediated silencing in *Amphimedon*, we identified the sponge homologs of *Drosophila* PRC2 components and found that the *Amphimedon* genome contains four copies of E(z) homologs, two copies of ESC homologs and one copy for each of the remaining components, SU(z)12 and Nurf55 (*Figure 4A*; *Figure 4—source data 1*).

PRC2 recruitment has been best characterised in *Drosophila* where PRC2 proteins repress their target genes by recruitment to Polycomb Response Elements (PREs), which contain binding sites for sequence-specific DNA binding proteins, including GAGA factor and members of the Krüppel-like factor family (*Müller and Kassis, 2006*; *Brown et al., 2005*; *Strutt et al., 1997*; *Simon and Kingston, 2009*; *Kassis and Brown, 2013*). To test whether *Amphimedon* PRC2 complexes might be recruited via a similar mechanism, we used the transcriptionally silenced regions marked by H3K27me3 in a *de novo* motif analysis (Materials and methods). We searched for short motifs (6–15 bp) on the basis that the known interaction sites of PREbinding proteins in *Drosophila* are of

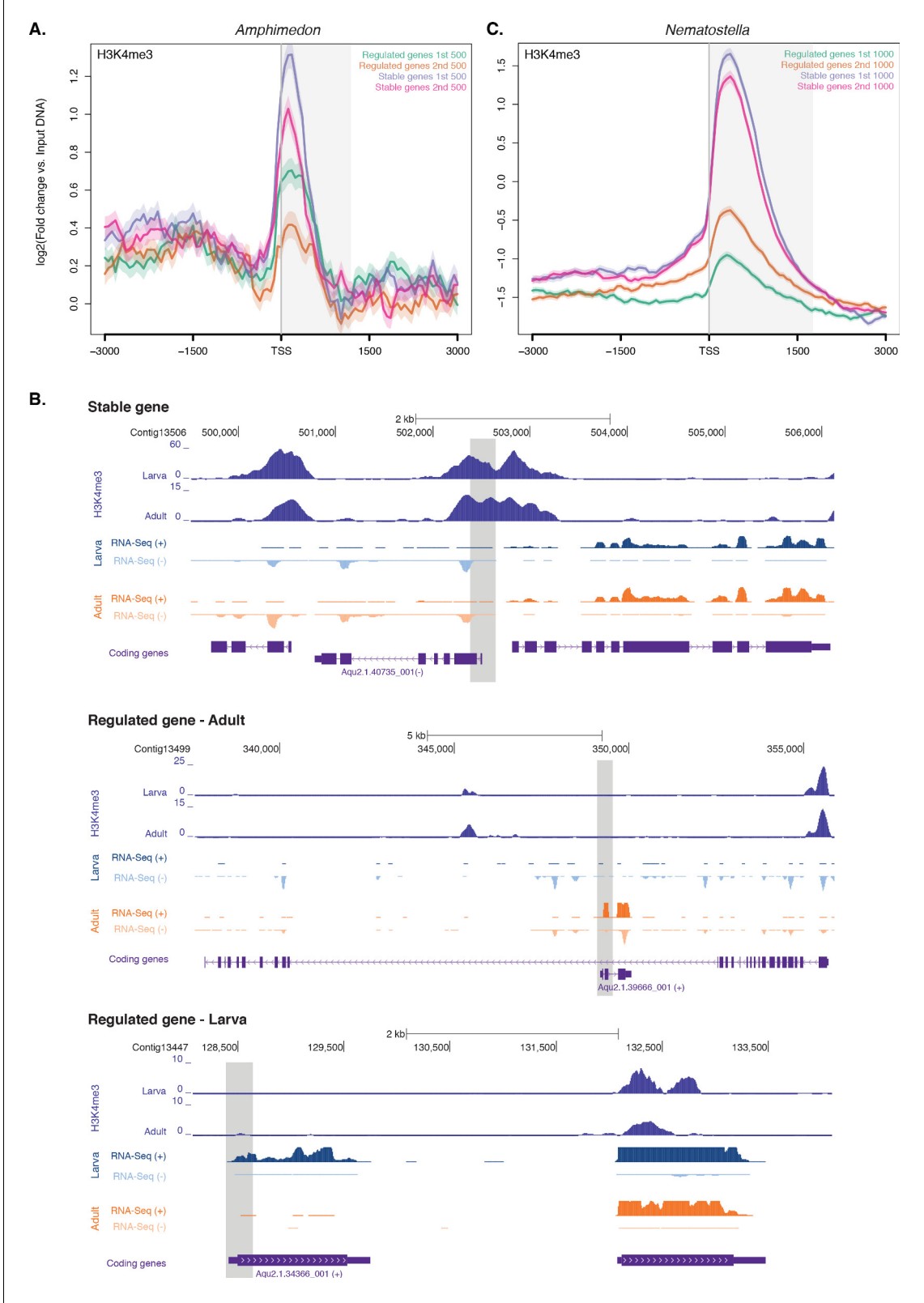

**Figure 3.** Expression without H3K4me3 in strongly developmentally regulated genes. (**A**) TSS-centred average input DNA normalised read coverage plot of H3K4me3 across 'regulated' and 'stable' protein-coding genes during *Amphimedon* development. The x-axis spans ± 3 kb around TSSs and represents the position within the gene relative to TSS. The y-axis represents the input DNA normalised enrichment for H3K4me3 ChIP-seq reads in adult *Amphimedon*. Light blue line: first 500 regulated genes. Orange line: second 500 regulated genes. Purple line: first 500 stable genes. Pink line:

*Figure 3 continued on next page*

*Figure 3 continued*

second 500 stable genes. The shaded gray area represents the average size of *Amphimedon* coding sequences. (B) Input DNA-normalized H3K4me3 coverage and RNA-seq expression in adult for Aqu2.1.40735_001, a gene stably expressed during *Amphimedon* development, Aqu2.1.39666_001, a regulated gene with adult-specific expression, and Aqu2.1.34366_001, a regulated gene with larva-specific expression. (C) TSS-centred average input DNA normalised read coverage plot of H3K4me3 across 'regulated' and 'stable' protein-coding genes during *Nematostella vectensis* development. The x-axis spans ± 3 kb around TSSs and represents the position within the gene relative to TSS. The y-axis represents the input DNA normalised enrichment for H3K4me3 ChIP-seq reads in *Nematostella* adult female polyps. The shaded gray area represents the average size of *Nematostella* coding sequences.

The following figure supplement is available for figure 3:

**Figure supplement 1.** ChIP-seq profiles of RNAPII, total histone H3, H3K36me3 and H3K27me3 across regulated and stable genes.

approximately this length (~8 bp). Conserved binding motifs similar to the GAGA and Krüppel-like factors, in addition to binding motifs similar to homeodomain-containing developmental regulators (*e.g.,* Irx family members), were significantly enriched (*E*-value<0.05) in the DNA associated with the H3K27me3 silenced regions in both adult and larva (***Figure 4B***; ***Figure 4—figure supplement 1***). As in eumetazoans, this result suggests that *Amphimedon* PRC2 complexes are likely to be recruited

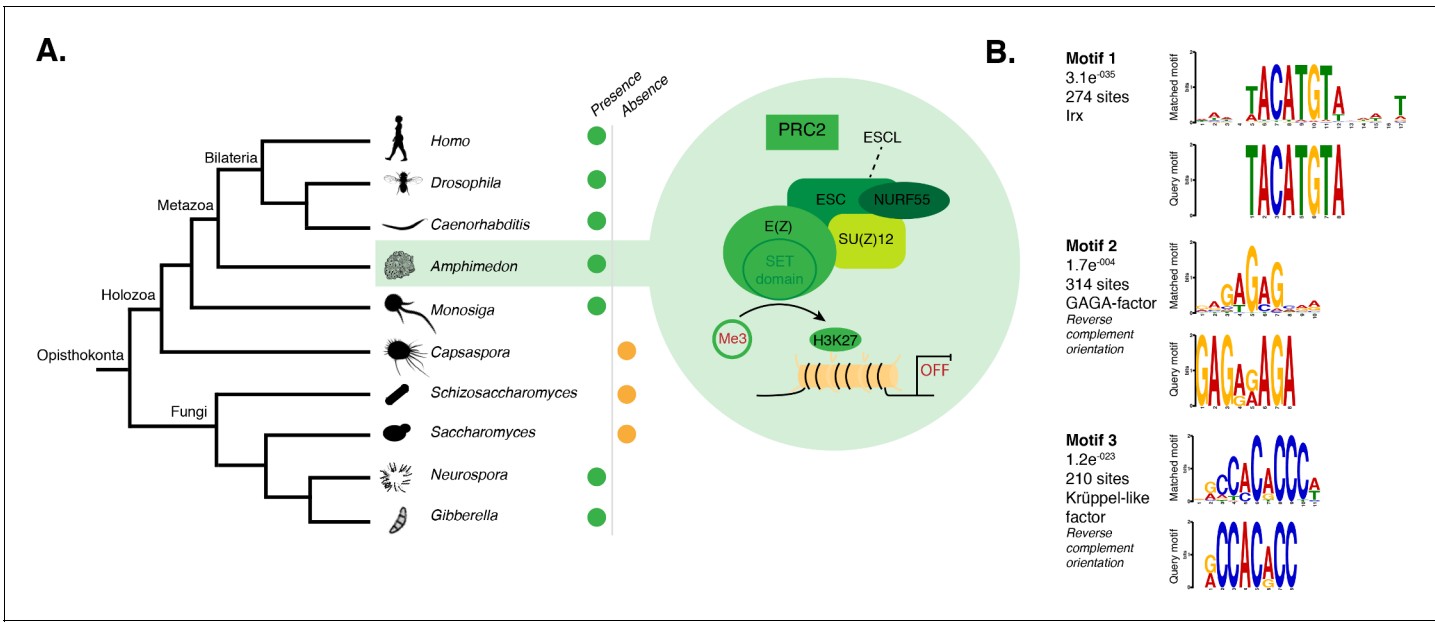

**Figure 4.** DNA motifs overrepresented in H3K27me3 transcriptionally silenced regions. (A) Diagram representing the composition of *Drosophila* PRC2 complex and its four core components: the catalytic subunit of the complex E(z), the zinc finger protein SU(z)12, the WD-repeat protein ESC and the histone-binding protein Nurf55. E(z) is responsible for the main enzymatic activity of PRC2, which is to trimethylate histone H3 at lysine 27, yielding H3K27me3. Adapted from (***Vissers et al., 2012***). The presence (green) or absence (orange) of PRC2 and its core components in the different opisthokont species represented in the phylogenetic tree (left) is shown. *Amphimedon* is highlighted in green. (B) Sequence logos of a subset of the DNA motifs determined by MEME-ChIP analysis to be significantly enriched in the transcriptionally silenced regions marked by H3K27me3 in adult *Amphimedon*. For each motif, the best TOMTOM match to a motif in the JASPAR CORE and UniPROBE mouse databases, the *E*-value and the number of sites contributing to the construction of the motif are shown, respectively. The matched motif is shown on the top and the query motif is shown on the bottom.

The following source data and figure supplement are available for figure 4:

**Source data 1.** Putative orthologs of *Drosophila* PcG components and associated factors in yeast, *Capsaspora*, sponge, nematode, and human genome.

**Figure supplement 1.** Matching sequence logos of the DNA motifs determined by MEME-ChIP analysis to be significantly enriched in the transcriptionally silenced regions marked by H3K27me3 in both adult and larva.

through PRE-like sequences and may target developmental regulators for H3K27me3 deposition and transcriptional silencing (*Margueron and Reinberg, 2011*; *Di Croce and Helin, 2013*; *Boyer et al., 2006*).

## Subset of *Amphimedon* lincRNAs is associated with an enhancer-like chromatin state

An additional layer of regulatory complexity in eumetazoan development is provided by long intergenic non-coding RNAs (lincRNAs) (*Ulitsky, 2016*; *Hezroni et al., 2015*; *Quinn and Chang, 2016*), which have been recently demonstrated to be developmentally expressed in sponges (*Gaiti et al., 2015*; *Bråte et al., 2015*). Here, we extended these analyses and analyzed the chromatin states of *Amphimedon* long intergenic ncRNAs (lincRNAs) (*Gaiti et al., 2015*), avoiding lncRNAs in protein-coding sequence introns or antisense to coding genes, which may yield ambiguous signals.

Previous studies have shown that the ratio of H3K4me1-to-H3K4me3 marks around TSSs can separate lincRNAs into enhancer-like lincRNAs (elincRNAs; high H3K4me1-to-H3K4me3 ratio) and canonical promoter-like lincRNAs (plincRNAs; low H3K4me1-to-H3K4me3 ratio) (*Sebé-Pedrós et al., 2016*; *Marques et al., 2013*; *Ilott et al., 2014*). Thus, to explore whether sponge lincRNAs might originate from enhancer regions, we interrogated our ChIP-seq data sets and calculated the relative ratio of H3K4me1-to-H3K4me3 in a 4 kb window centered on lincRNA TSSs. Only lincRNAs in scaffolds larger than 10 kb that overlapped with regions of enrichment of H3K4me1, H3K4me3, and RNAPII were used in this analysis (n = 217). Similarly to *Ilott et al. (2014)*, we arbitrarily adopted a H3K4me1-to-H3K4me3 ratio of >1.2 and <0.8 to define elincRNAs and plincRNAs, respectively. Based on these criteria, we found 153 putative elincRNAs (70%) significantly enriched for H3K4me1 over H3K4me3 (Mann-Whitney U test, p-value=2.272e-05) and 21 (10%) putative plincRNAs with canonical promoter signature, *i.e.*, higher enrichment of H3K4me3 over H3K4me1 (Mann-Whitney U test, p-value=1.925e-07). 43 (20%) lincRNAs could not be assigned to either group, that is, 0.8 < H3K4me1-to-H3K4me3 < 1.2 (*Figure 5A–D*; *Figure 5—source data 1*; *Figure 5—figure supplement 1*).

These results indicate that sponge lincRNAs can be separated in two distinct populations of poly (A)$^+$ transcripts based on the chromatin status at their TSSs. Although these two populations resemble those found in human, mouse and *Capsaspora* lincRNAs (*Sebé-Pedrós et al., 2016*; *Marques et al., 2013*; *Ilott et al., 2014*), their functional significance is yet to be determined.

## Identification of enhancer elements in *Amphimedon*

To identify putative enhancer elements in *Amphimedon in silico*, we selected distal H3K4me1 regions of enrichment (high confidence regions, representing reproducible events across true biological replicates) that did not overlap TSSs (±200 bp) of protein-coding genes and lncRNAs, but overlapped with regions designated as being in an enhancer chromatin state based on the ChromHMM analysis ('TxEnhA' or 'EnhWk' or 'EnhP' state in adult; 'TxEnhA1' or 'TxEnhA2' or 'EnhWk' or 'EnhP' state in larva, which consist of typical eumetazoan enhancer histone H3 PTM patterns) (*Figure 6A*). A subset of these regions was also marked by H3K27ac, and therefore likely to be transcriptionally active (*Figure 6A and B*; *Figure 6—source data 1*). These predicted activated enhancer-like regions showed a significant enrichment of H3K4me1 and H3K27ac over H3K4me3 (Mann-Whitney U test, p-value<2.2e-16; *Figure 6C*; *Figure 6—figure supplement 1*), a biochemical signature typical of eumetazoan enhancers (*Schwaiger et al., 2014*). Interestingly, RNAPII occupied some of these *Amphimedon* predicted activated enhancer-like elements (35% and 41% in adult and larva, respectively), suggesting poly(A)+ enhancer RNAs could be transcribed from these regions (*Natoli and Andrau, 2012*; *Li et al., 2016*; *Kim et al., 2010*) (*Figure 6A–D*; *Figure 6—figure supplement 2*). Alternatively, but not exclusively, this might represent the result of chromatin looping and the simultaneous pulldown of both enhancers and promoters with the RNAPII antibody (*Shlyueva et al., 2014*).

In eumetazoans, genes encoding transcriptional regulators are themselves regulated by multiple enhancer elements (*Schwaiger et al., 2014*; *Shlyueva et al., 2014*; *Nègre et al., 2011*; *Bogdanovic et al., 2012*; *Woolfe et al., 2005*; *Heintzman et al., 2009*). We therefore performed *de novo* motif analysis and, despite the limited power of motif detection due to the inherent cellular heterogeneity of our starting material, we were able to show that consensus binding motifs of key

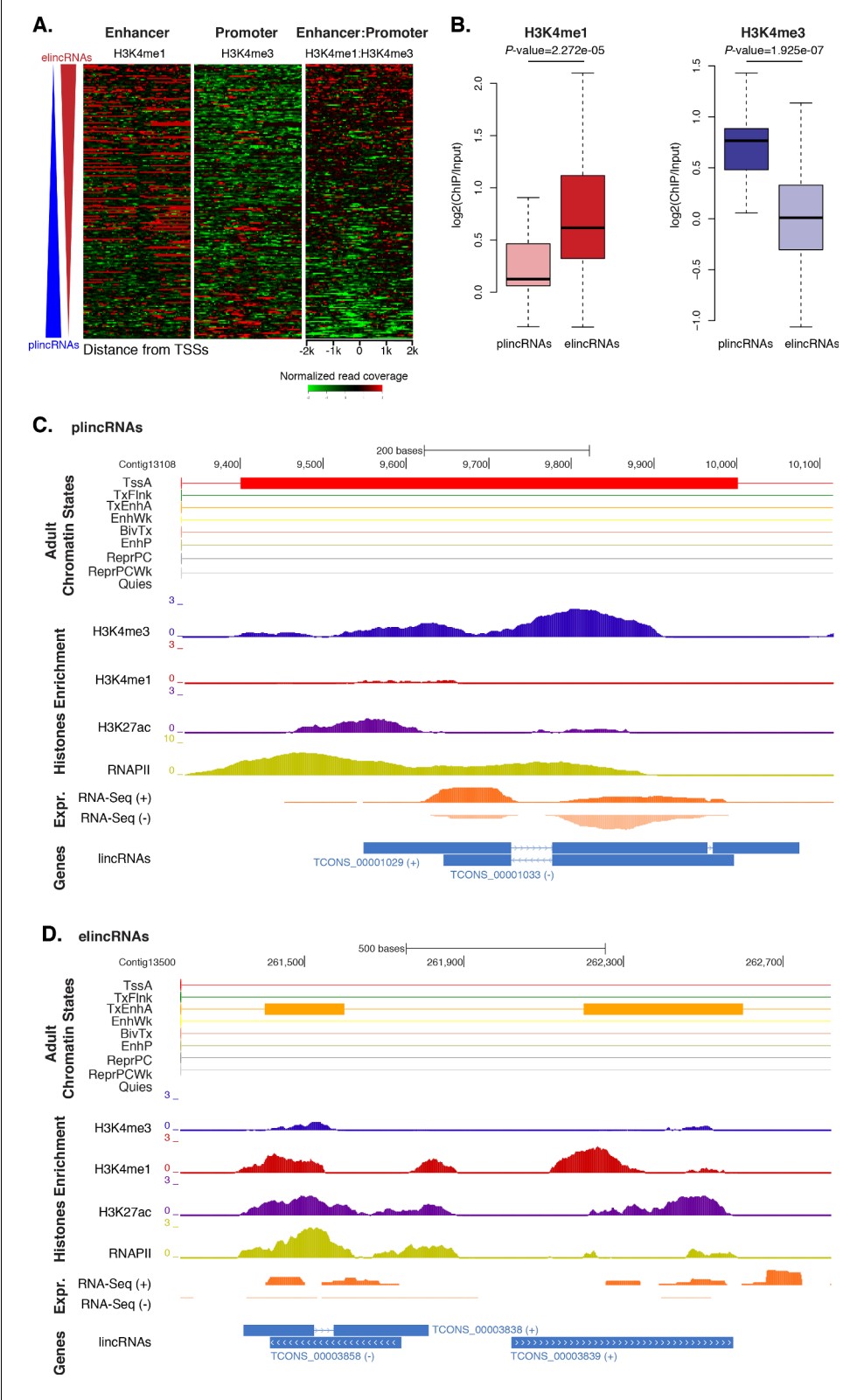

**Figure 5.** *Amphimedon* lincRNA populations defined by histone PTM signatures. (**A**) Heatmap showing the average read normalised coverage of H3K4me1, H3K4me3 and their ratio in adult *Amphimedon* across a 4 kb interval centred on TSSs of lincRNAs. Each line of the heatmaps represents a single lincRNA (y-axis). Profiles are sorted based on the differences in enrichment between H3K4me1 and input DNA, and H3K4me3 and input DNA, respectively. Also provided is the H3K4me1:H3K4me3 log2(ratio) around TSSs. (**B**) Enrichment of H3K4me1 (left) and H3K4me3 (right) (ChIP versus input)

*Figure 5 continued on next page*

*Figure 5 continued*

at plincRNAs and elincRNAs. *P*-values are indicated for Mann-Whitney U test. (**C**) Example of lincRNAs with promoter-like chromatin signature (plincRNAs). For the definition of adult chromatin states see *Figure 1A*. Promoter-like lincRNAs (blue) are shown, along with input DNA-normalized coverage of different histone modifications and RNA-seq expression in adult. (**D**) Same as (**C**) but for lincRNAs with enhancer-like chromatin signature (elincRNAs).

The following source data and figure supplement are available for figure 5:

**Source data 1.** Annotation of putative elincRNAs and plincRNAs.
**Figure supplement 1.** Additional examples of plincRNAs and elincRNAs.

developmental transcription factor families were over-represented in the adult predicted activated enhancer-like sequences, including Zinc finger, Irx, SOX and POU binding motifs (*Figure 6E*; *Figure 6—figure supplement 3*). It is noteworthy that Zinc fingers can also be involved in roles that might be unrelated to directly regulating gene expression per se, *e.g.*, chromatin remodeling (*Wysocka et al., 2006*). Similar binding motifs were obtained analysing the larva predicted activated enhancer-like sequences (*Figure 6—figure supplement 4*). Next, we examined whether the sponge predicted activated enhancer-like elements were preferentially located next to protein-coding genes involved in development and/or transcriptional regulation. By searching for the closest located TSSs to each of the predicted activated enhancer-like elements in *Amphimedon*, we nominated putative target protein-coding genes. Akin to eumetazoans, these nearest neighbor genes were significantly enriched for Gene Ontology (GO) terms associated with transcription factor activity and developmental processes (Hypergeometric test, FDR adjusted p-value<0.01) (*Figure 6F*), and comprised several transcription factors, including SOX2, FOS and NF-kB (*Figure 6D*; *Figure 6—figure supplement 5*; *Figure 6—source data 2–5*).

Vertebrates exhibit expansive intergenic regions where the majority of predicted enhancers are located (*ENCODE Project Consortium, 2012*; *Djebali et al., 2012*). In contrast, in *Amphimedon*, which has a highly compact genome with minimal intergenic regions (*Fernandez-Valverde and Degnan, 2016*), predicted activated enhancer-like elements were predominantly intragenic, with only a minority found in intergenic regions (9% and 20% in adult and larva, respectively) (*Fernandez-Valverde and Degnan, 2016*). This, along with the strong enrichment of chromatin states typically associated with eumetazoan enhancers – 'TxEnhA' and 'EnhWk' – in introns (*Figure 1A*; *Figure 1—figure supplement 4*), suggests a similar overall genomic distribution between *Amphimedon*, *Nematostella* and *Drosophila* enhancer elements (*Schwaiger et al., 2014*; *Nègre et al., 2011*; *Arnold et al., 2013*).

Greater intron length often associates with the presence of highly conserved non-coding elements (*Irimia et al., 2011*). We, therefore, extracted the introns that harbour predicted activated enhancer-like elements and compared their size distribution to the size of all intronic regions found across the genome. The former were significantly longer than the average genomic intron size, with a mean of 332 bp and 256 bp, and a median of 99 bp and 71 bp, respectively (Ansari-Bradley test, p-value=0.06151; Mann-Whitney U test, p-value=1.927e-06) (*Figure 6G*), suggesting that a *cis*-regulatory expansion appear to have occurred primarily in intronic rather than intergenic regions in *Amphimedon*.

### *Cis*-regulation constrains genome architectures over 700 Myr of evolution

Highly conserved non-coding regulatory elements are often associated not only with greater intron length, but also with genes encoding developmental regulators (*Woolfe et al., 2005*; *Vavouri et al., 2007*; *Sandelin et al., 2004*). Particularly interesting are the conserved ancestral microsyntenic pairs (herein microsyntenic units) that consist of either (i) two neighbor genes that share common *cis*-regulatory elements, or (ii) a developmental regulator and nearby functionally unrelated gene(s), whose introns harbor conserved *cis*-regulatory elements (*Kikuta et al., 2007*; *Irimia et al., 2013*; *Engström et al., 2007*; *Irimia et al., 2012*; *Naville et al., 2015*). Experimental

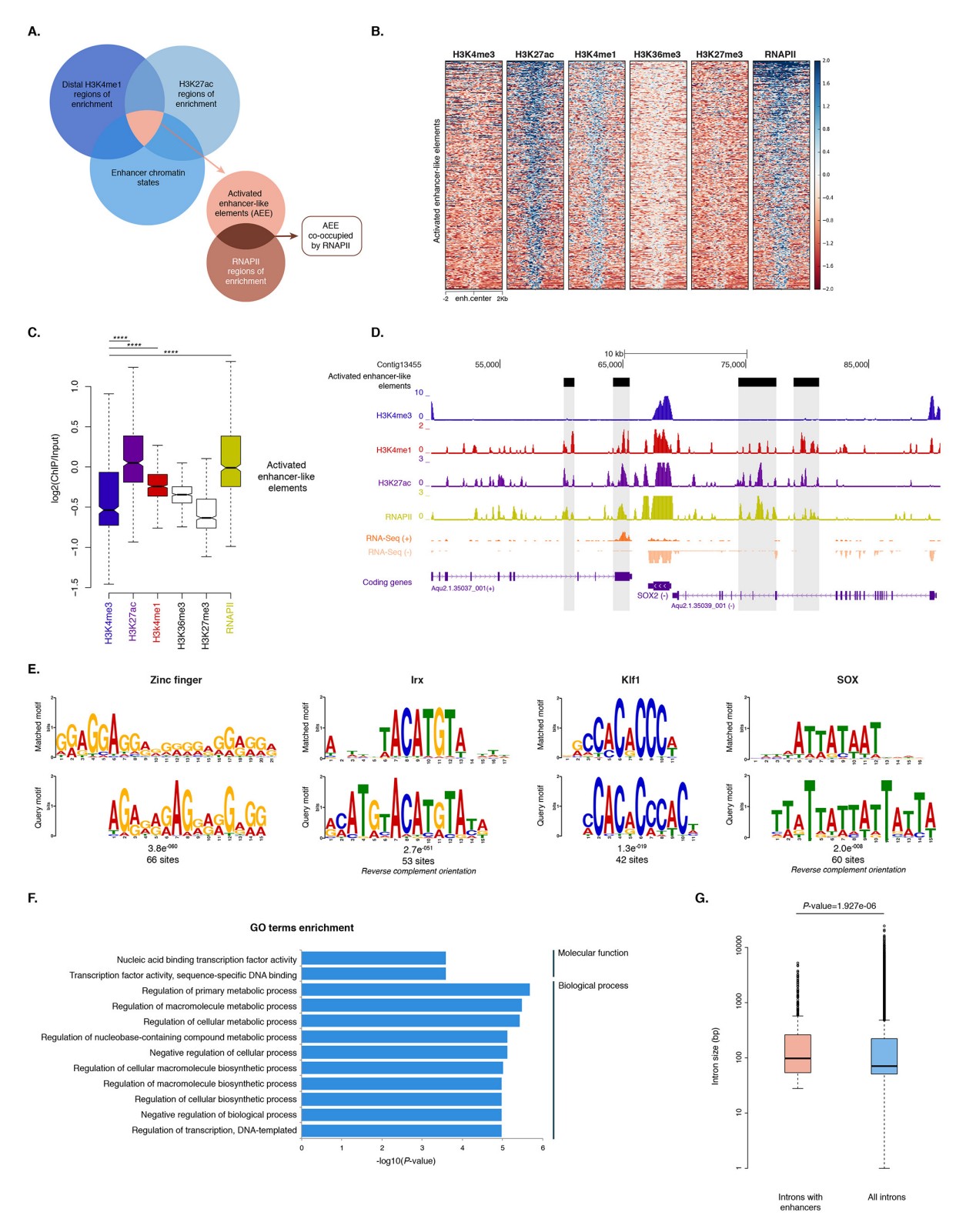

**Figure 6.** Distal enhancer regulation at the dawn of animals. (A) Overview of the computational filtering pipeline adopted to predict the putative *Amphimedon* activated enhancer-like elements. See main text and Materials and methods for details. (B) Heatmap showing different histone modifications enrichment at predicted activated enhancer-like elements (±2 kb of flanking regions). (C) Boxplot showing enrichment of different histone modifications (ChIP versus input) at predicted activated enhancer-like elements, showing that activated enhancer-like elements have higher H3K4me1

*Figure 6 continued on next page*

*Figure 6 continued*

than H3K4me3 levels, a typical characteristic of eumetazoan enhancers. Four asterisks (****) indicate p-values<2.2e-16 for Mann-Whitney U test between H3K4me3 and H3K27ac, between H3K4me3 and H3K4me1, and between H3K4me3 and RNAPII, respectively. (D) Example of predicted activated enhancer-like elements. Protein coding genes (purple) are shown, along with input DNA-normalized coverage of different histone modifications and RNA-seq expression in adult. Regions of enrichments (high confidence peaks, representing reproducible events across true biological replicates) corresponding to the predicted activated enhancer-like elements are highlighted in grey. (E) Sequence logos of the DNA motifs determined by MEME-ChIP analysis enriched in the adult predicted activated enhancer-like sequences. For each motif, the best match to a motif in the JASPAR CORE and UniPROBE mouse databases, the *E*-value and the number of sites contributing to the construction of the motif are shown, respectively. The matched motif is shown on the top and the query motif is shown on the bottom. (F) Gene Ontology (GO) enrichment activities of the nearest neighbor protein-coding genes of the adult predicted activated enhancer-like elements are shown. Bar length indicates the significance of the enrichment (Hypergeometric test; -log10[adjusted *P*- value]). Only the top ten GO biological process terms are shown. See *Figure 6—source data 2* for the complete list. (G) Boxplot showing the size of introns that harbour adult activated enhancer-like elements versus all introns in the genome. The y-axis indicates the intron size (bp) in log scale. *P*-value is indicated for Mann–Whitney U test.

The following source data and figure supplements are available for figure 6:

**Source data 1.** Genomic location of all the predicted activated enhancer-like elements and their distance to the closest TSS.

**Source data 2.** Functional annotation of nearest neighbors genes of the adult predicted activated enhancer-like elements.

**Source data 3.** Functional annotation of nearest neighbors genes of the larva predicted activated enhancer-like elements.

**Source data 4.** GO term enrichment outcome for the nearest neighbors genes of the adult predicted activated enhancer-like elements (Hypergeometric test, FDR<0.01).

**Source data 5.** GO term enrichment outcome for the nearest neighbors genes of the larva predicted activated enhancer-like elements (Hypergeometric test, FDR<0.01).

**Figure supplement 1.** Activated enhancer-like elements have higher H3K4me1 than H3K4me3 levels.

**Figure supplement 2.** Examples of CEL-seq or RNA-seq expression detected at putative activated enhancer-like sites, suggesting that 1D eRNAs, which are generally polyadenylated (*Natoli and Andrau, 2012*; *Li et al., 2016*), might be transcribed from these regions.

**Figure supplement 3.** Additional sequence logos of the DNA motifs determined by MEME-ChIP analysis to be significantly enriched in the adult predicted activated enhancer-like sequences.

**Figure supplement 4.** Matching sequence logos of the DNA motifs determined by MEME-ChIP analysis to be significantly enriched in the predicted activated enhancer-like sequences in both adult and larva.

**Figure supplement 5.** Examples of predicted enhancer-like elements in proximity of well-known developmental and transcription factor genes.

evidence has been provided for the existence of this type of *cis*-regulation in vertebrates (*Irimia et al., 2012*; *Naville et al., 2015*).

To test whether this is an ancient *cis*-regulatory mechanism maintained through animal evolution, we assessed the spatial relationship between the genes of each of the 80 microsyntenic units previously reported to be present in the *Amphimedon* genome (*Irimia et al., 2012*) and clarified their orthology, confirming the presence of 60 unambiguous microsyntenic units. Remarkably, 43 of these 60 evolutionary conserved metazoan microsyntenies contained putative enhancer-like signatures in *Amphimedon* adults (*Figure 7A*; *Figure 7—source data 1*; *Figure 7—figure supplement 1*). This was a much higher fraction relative to a control set consisting of 60 pairs of two randomly selected nonsyntenic neighbor genes (1,000 iterations; p-value<0.00001). This pattern was substantiated by the finding of larva enhancer-like signatures in 16 of the 60 microsyntenic units, seven of which contained both larva and adult predicted enhancer-like elements (*Figure 7A*; *Figure 7—source data 1*; *Figure 7—figure supplement 1*).

A striking case of conserved gene linkage involves the Islet LIM homeobox gene (*Isl*), which plays conserved roles in animal development (*Thor and Thomas, 1997*; *Liang et al., 2011*), and *Scaper* (S-phase cyclin A-associated protein in the ER) (*Figure 7B*). The *Amphimedon Scaper* contains 25

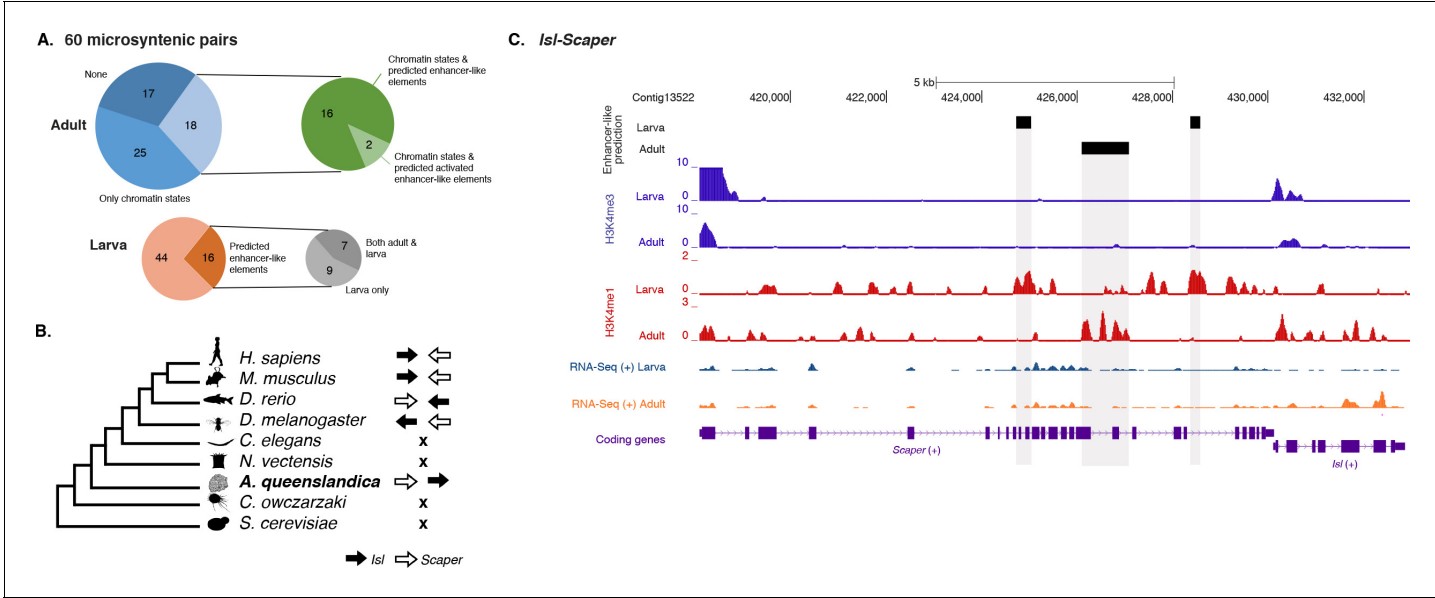

**Figure 7.** *Amphimedon* enhancer-like elements are enriched in metazoan-specific microsyntenic units. (A) Putative adult and larva enhancer-like signatures identified in the 60 metazoan-specific microsyntenic pairs investigated. (B) The cladogram represents known phylogenetic distribution of the *Isl2-Scaper* microsyntenic gene pair across opisthokonts. The orientation of the arrow corresponds to gene orientation. *Isl2-Scaper* is not conserved in yeast, *Capsaspora*, *Nematostella* and *C. elegans*. (C) Enhancer elements in the *Isl-Scaper* microsyntenic gene pair locus in *Amphimedon*. *Scaper* and *Isl* genes (purple) are shown, along with input DNA-normalized coverage of H3K4me3 and H3K4me1 and RNA-seq expression in both adult and larva. Regions of enrichments (high confidence peaks, representing reproducible events across true biological replicates) corresponding to the predicted enhancer-like elements located within the introns of *Scaper* are highlighted in grey.

The following source data and figure supplement are available for figure 7:

**Source data 1.** 60 microsyntenic units representing functional gene linkages and presence-absence of chromatin states containing typical eumetazoan enhancer histone PTM patterns ('EnhP',' EnhWk',' TxEnhA') (adult only) and/or *in silico* predicted enhancer-like elements (both larva and adult).

**Figure supplement 1.** Additional examples of predicted enhancer-like elements in conserved microsyntenic units.

introns, some of which are considerably longer (>1 kb) than the mean intron size (*Fernandez-Valverde et al., 2015*), and predicted enhancer-like elements located within its intron 10, 17 and 21 (*Figure 7C*). Likewise, the microsynteny of *Tfap4* (transcription factor AP-4) (*Simionato et al., 2007*) and *Glis2* (GLIS family zinc finger 2) is deeply conserved. Similar to an observation in vertebrates (*Abbasi et al., 2007*), the sponge *Glis2* contains two introns, of which the second harbors several adult predicted activated enhancer-like elements (*Figure 7—source data 1*; *Figure 7—figure supplement 1*). Together, these results suggest that the genomic location of some *cis*-regulatory elements likely places constraints on the evolution of nearby genes, leading to the occurrence of conserved microsyntenic gene blocks across the animal kingdom.

## Discussion

Since their point of divergence over 700 Mya, sponges and eumetazoans have had radically different evolutionary histories, with the eumetazoan ancestor giving rise to a range of morphologically-complex body plans, and the sponge ancestor yielding one basic morphologically-simple body plan. As both these lineages share a remarkably similar repertoire of developmental gene families (*Srivastava et al., 2010*; *Larroux et al., 2008*; *Richards et al., 2008*), these different evolutionary trajectories have yet to be reconciled in terms of genome content and organization. Recently it has been shown that, despite having a highly compact genome with minimal intergenic regions (*Fernandez-Valverde and Degnan, 2016*), *Amphimedon* displays dynamic developmental gene expression akin to eumetazoans (*Gaiti et al., 2015*; *Levin et al., 2016*). By generating the first, to our

knowledge, comprehensive genome-wide maps of histone H3 PTMs and putative enhancer elements in a non-eumetazoan animal, we determine that this transcriptional complexity is paralleled by regulatory complexity encoded by combinatorial histone H3 PTM patterns in this sponge.

## Histone H3 PTMs have conserved role(s) in the modulation of gene expression across metazoans

Despite *Amphimedon*'s morphological simplicity, we find strong evidence in this sponge for the existence of a range of regulatory states that underlie eumetazoan development. For instance, the genome-wide promoter analysis of H3K4me3 – the canonical and widespread eukaryotic histone H3 PTM of active transcription – reveals a complex correlation between H3K4me3-containing nucleosome occupancy and gene expression in *Amphimedon* adults and larvae, consistent with an active and finely tuned role for H3K4me3 in modulating transcriptional activity and expression variability of developmental genes. Unexpectedly, we identify a small subpopulation of highly and specifically expressed genes that challenge this premise and are transcribed in the absence of H3K4me3 in *Amphimedon* and *Nematostella*. This subpopulation of genes differs from most other developmentally-expressed genes that possess the H3K4me3 mark, in having much more stage-restricted expression profiles; in this analysis, most are expressed only in one stage of development. Although it could be argued that this apparent absence of H3K4me3 is the consequence of the expression of regulated genes being spatially confined to specific cell populations, thus potentially limiting our detection sensitivity with our cell admixture ChIP-seq, these results run parallel to the recent finding by *Pérez-Lluch et al. (2015a)* that *Drosophila* and *C. elegans* exhibit the same pattern, suggesting that this newly-discovered feature is conserved across the animal kingdom. As the expression of the developmentally regulated genes is required only for a limited period, the absence of H3K4me3 mark would allow their rapid on-off switching. Alternative mechanisms, such as the transient binding of transcription factors, appear to play a major role in regulating the expression of these genes (*Pérez-Lluch et al., 2015a*, *2015b*).

## Model of an evolutionarily conserved mechanism of PRC2-mediated gene silencing

Polycomb Repressive Complex 2 (PRC2) primarily trimethylates histone H3 on lysine 27 and has been conserved throughout opisthokonts evolution, with its core subunits (E(z), SU(z)12, ESC and Nurf55) being present in animals, choanoflagellates and multicellular fungi, but absent in *Capsaspora*, and budding and fission yeast (*Sebé-Pedrós et al., 2016*; *Margueron and Reinberg, 2011*; *Shaver et al., 2010*; *Jamieson et al., 2013*; *Connolly et al., 2013*; *Ikeuchi et al., 2015*; *Whitcomb et al., 2007*) (*Figure 4A*). This is consistent with PRC2 complex being lost in several unicellular lineages. One of the ancestral roles of PCR2 in opisthokonts may have been in defense response against viruses and transposable elements, or insertion of new genes (*Jamieson et al., 2013*), prior to being co-opted for cell-type specific developmental regulation in animals, where H3K27me3 and PRC2 are required for transmitting the memory of repression across generations and during development (*Margueron and Reinberg, 2011*; *Shaver et al., 2010*; *Gaydos et al., 2014*; *Barski et al., 2007*). In fact, PRC2 often regulates deposition of H3K27me3 marks at loci encoding developmental regulators (*Ha et al., 2011*; *Margueron and Reinberg, 2011*; *Barski et al., 2007*). The finding of short conserved developmental transcription factor-binding-sites in *Amphimedon* H3K27me3 silenced regions is consistent with this evolutionary scenario. Analogous to recent findings in plants (*Deng et al., 2013*; *Hecker et al., 2015*), the identification of an enriched motif in the H3K27me3 silenced regions similar to the GAGA factor binding site, a component of the *Drosophila* Polycomb group response elements, suggests a role for the GAGA factor binding sites in strengthening PRC2 recruitment to target genes (*Müller and Kassis, 2006*; *Simon and Kingston, 2009*; *Kassis and Brown, 2013*). It is noteworthy that a sponge homolog of *Drosophila* GAGA factor was not identified in the current *Amphimedon* genome assembly (*Figure 4—source data 1*), suggesting the convergent co-option of other DNA binding proteins with analogous role(s) in the recruitment of PRC2.

## The origin of animal distal enhancer regulation

Analysis of *cis*-regulatory DNA and histone PTMs have revealed that some *cis*-regulatory mechanisms, such as those associated with proximal promoters, are present in non-animal holozoans, while others appear to have evolved later on the stem leading to the crown metazoans, most notably distal enhancers (*Sebé-Pedrós et al., 2016*; *Schwaiger et al., 2014*). The latter has been posited to be one of the key contributing factors underlying the spatial and temporal coordination of cell differentiation that defines animal development (*Levine, 2010*; *Levine et al., 2014*; *Levine and Tjian, 2003*; *Peter and Davidson, 2011*). Our *in silico* prediction of *Amphimedon* enhancer elements based on histone H3 PTM co-localization patterns is consistent with these elements evolving along the metazoan stem at the transition to multicellularity (*Sebé-Pedrós et al., 2016*). Interestingly, promoter DNA regulatory elements to allow for context and cell type-specific gene expression also appeared to evolve in stem metazoans (*Fernandez-Valverde and Degnan, 2016*), suggesting these are also a critical component of the animal *cis*-regulatory landscape. *Amphimedon* predicted enhancer-like elements are characterized by the same combination of histone H3 PTMs as in eumetazoans, which appear to be lacking in unicellular holozoan relatives of animals (*Sebé-Pedrós et al., 2016*; *Bulger and Groudine, 2011*). Their preferential association with developmental and transcriptional regulators suggests that *Amphimedon* enhancer elements are likely to regulate developmental genes in a manner akin to eumetazoans (*Schwaiger et al., 2014*; *Shlyueva et al., 2014*; *Nègre et al., 2011*; *Bogdanovic et al., 2012*; *Woolfe et al., 2005*; *Heintzman et al., 2009*). Enhancer elements are known to be associated with the transcription of both short poly(A)$^-$ and long poly(A)$^+$ enhancer RNAs (2D and 1D eRNAs, respectively) (*Natoli and Andrau, 2012*; *Li et al., 2016*; *Kim et al., 2010*). The presence of RNAPII and the detection of expression at a subset of the *Amphimedon* activated enhancer-like elements is consistent with this notion (*Figure 6—figure supplement 2*). Although non-coding transcription at these enhancers will need to be investigated in detail, this co-occupancy of enhancer elements and RNAPII has also been observed in *Nematostella* and bilaterians (*Schwaiger et al., 2014*; *Li et al., 2016*; *Kim et al., 2010*; *De Santa et al., 2010*; *Chen et al., 2013*), where these elements might be physically interacting with the transcription initiation complex at the TSS of their target gene(s) (*Schwaiger et al., 2014*).

Unlike bilaterians, where the transcriptional repressor CCCTC-binding factor (CTCF) localizes with cohesin genome-wide and is involved in enhancer-promoter long-range interactions and higher-order chromatin structure (*Lee and Iyer, 2012*; *Seitan et al., 2013*; *Merkenschlager and Odom, 2013*), *Amphimedon* lacks CTCF (*Heger et al., 2012*). This likely constrains *Amphimedon* enhancer interactions with the proximal promoter transcriptional machinery to short distances. Chromatin looping of enhancers to their target promoters in this sponge might therefore occur through a CTCF-independent cohesin binding mechanism, as proposed in cnidarians, which also lack CTCF (*Schwaiger et al., 2014*). Alternatively, but not exclusively, RNAPII and its associated transcriptional machinery may track through the intervening DNA between enhancers and promoters (*Li et al., 2016*), and might be the preferred mechanism of enhancer-promoter interactions in this sponge. The co-occupancy of *Amphimedon* enhancer-like elements and RNAPII supports this mechanism of transcriptional activation. Future studies of the 3D genome architecture will be crucial in elucidating the mechanism of enhancer-promoter interaction in this sponge and other early-branching non-bilaterian animals lacking this architectural protein (*Gaiti et al., 2016*).

Finally, we find strong evidence for *cis*-regulatory elements being important for the maintenance of metazoan-specific microsyntenic gene blocks over 700 Myr of evolution. The emergence of distal enhancer regulation prior to metazoan cladogenesis could explain the pervasiveness of conserved syntenic regulatory blocks in animal genomes and the absence of these blocks in their unicellular relatives (*Srivastava et al., 2010*; *Sebé-Pedrós et al., 2016*; *Irimia et al., 2013*, *2012*; *Bulger and Groudine, 2011*; *Putnam et al., 2007*; *Duan et al., 2010*). The strong evidence for enhancer elements being enriched in deeply conserved metazoan-specific microsyntenic units suggests that their genomic location is likely to constraint genome architecture, leading to the occurrence of conserved microsyntenies across the animal kingdom (*Irimia et al., 2013*, *2012*).

In conclusion, a conserved gene regulatory landscape similar to that of morphologically-complex eumetazoans appears to have been already in place at the dawn of animals, and thus likely to have originated at least 700 Mya. Specifically, there appears to have been fundamental changes in the *cis*-regulatory architecture of the genome along the metazoan stem, concomitant with the evolution

of animal multicellularity, including the apparent origin of distal enhancers and promoter types for cell-type-specificity and developmental regulation. With this in mind, we propose an evolutionary scenario in which *quantitative* rather than *qualitative* differences in regulatory mechanisms likely drive the evolution and diversification of eumetazoan body plans (*Figure 8*).

## Materials and methods

### Animal collection

*Amphimedon queenslandica* adults and larvae were collected from Heron Island Reef, Great Barrier Reef, Queensland, Australia, and reared as previously described (*Leys et al., 2008*).

### Antibodies

We used a mouse monoclonal antibody against the unphosphorylated C-terminal repeat of RNA polymerase II (RRID:AB_492629) (clone 8WG16, #05–952, Merck Millipore, Billerica, MA), a rabbit polyclonal antibody against H3K4me3 (RRID:AB_1977252) (#07–473, Merck Millipore), a rabbit

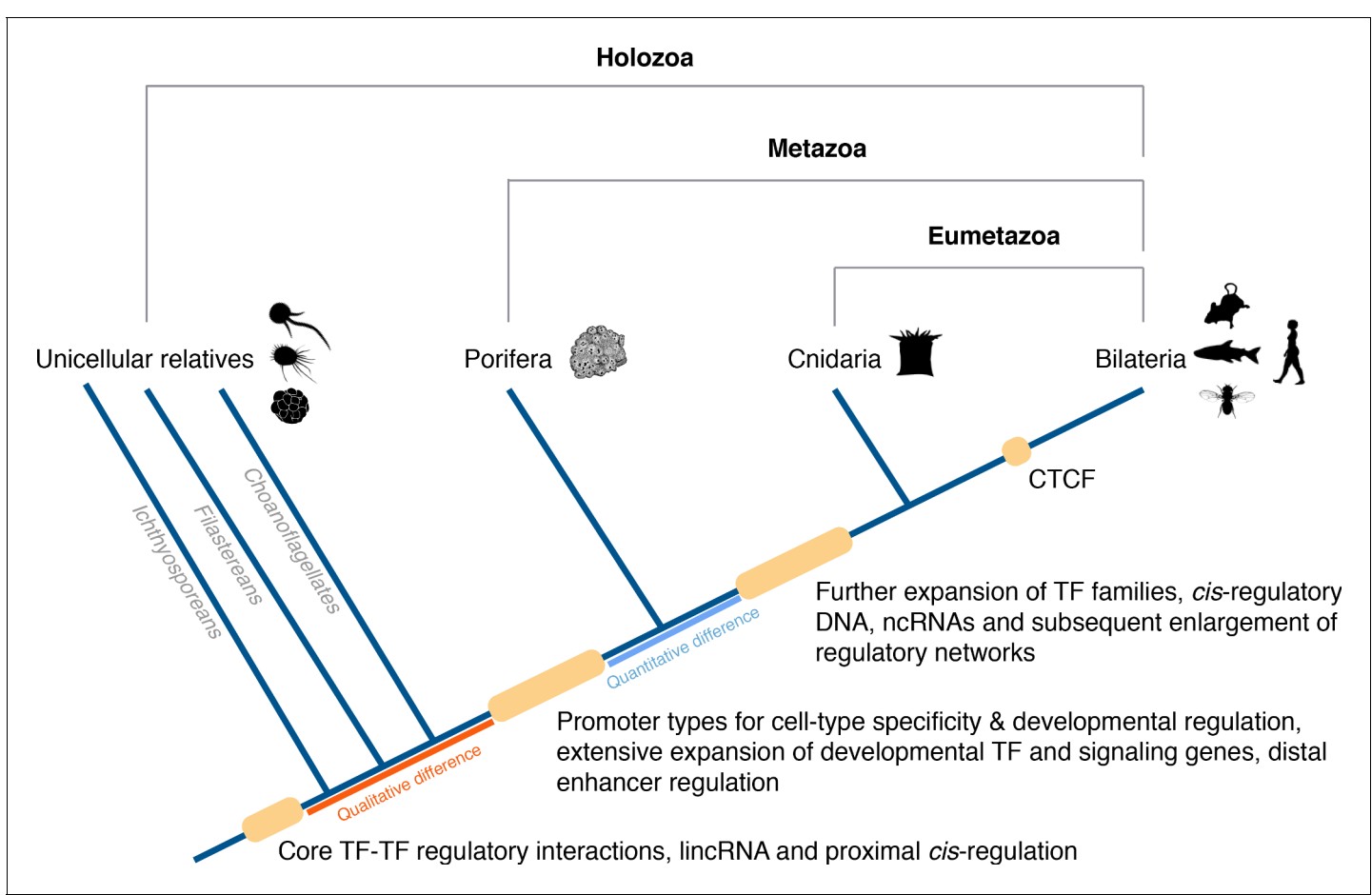

**Figure 8.** Origin of animal *cis*-regulatory complexity. The phylogenetic relationship of representative animal lineages and unicellular holozoans is shown here. Highlighted are the major genomic innovations that correlate with the emergence and early diversification of animals. Some components of the metazoan regulatory landscape may predate the split of the metazoan and holozoan lineages, including core TF-TF regulatory interactions and long intergenic non-coding RNAs, which have been recently identified in unicellular relatives of animals (*Sebé-Pedrós et al., 2016*; *de Mendoza et al., 2015*) but for which the evolutionary origin is still unclear. With a complex gene regulatory landscape already in place at the dawn of animals, the expansion of developmental gene families (encoding transcription factors and components of signaling pathways), *cis*-regulatory DNA and non-coding RNAs, along with the emergence of the architectural protein CTCF to allow more complex enhancer-promoter interactions, appear to underlie the evolutionary diversification of the eumetazoan body plans.

polyclonal antibody against H3K27me3 (RRID:AB_310624) (#07–449, Merck Millipore), a mouse monoclonal antibody against H3K4me1 (RRID:AB_10806625) (#17–676, Merck Millipore), a rabbit polyclonal antibody against H3K27ac (RRID:AB_310550) (#07–360, Merck Millipore), a rabbit monoclonal antibody against H3K36me3 (RRID:AB_10615601) (#17–10032, Merck Millipore), and a rabbit polyclonal antibody against histone H3 (RRID:AB_417398) (#07–690, Merck Millipore) (*Figure 1— source data 1*). The entire amino acid sequence of histone H3 is perfectly conserved between *Amphimedon* and other eukaryotes where these antibodies have been used successfully (*Sebé- Pedrós et al., 2016*; *Ercan et al., 2009*; *Barraza et al., 2015*; *Harmeyer et al., 2015*; *Liu et al., 2007*; *Eckalbar et al., 2016*) (*Figure 1—figure supplement 1*).

## Chromatin immunoprecipitation (ChIP) assays

Approximately a cm$^3$ of adult sponge tissue was squeezed through a fine cloth and cells ($\sim10^7$) were crosslinked in 2% formaldehyde for 5 min at room temperature (RT). Larvae ($\sim$350) were pooled, homogenized and crosslinked as above. A similar procedure was then adopted for both developmental stages. Specifically, crosslinking was quenched with 125 mM glycine for 5 min at RT. Cells were washed twice in 0.22 μm filtered seawater and centrifuged at 500 g for 5 min. Pelleted cells were lysed in SDS Lysis buffer (10 mM EDTA, 50 mM Tris-HCl at pH 8.0, 1% SDS, plus protease and phosphatase inhibitors), incubated for at least 10 min on ice, and sonicated for 12 min (12 cycles, each one 30 s 'ON', 30 s 'OFF') in a Bioruptor Sonicator (Diagenode, Seraing, Belgium) to generate 200–300 bp fragments. Optimal sonication conditions were previously determined by testing a range of sonication cycles (from 5 to 30); 12 cycles were deemed as optimal. Non-soluble material was removed from the lysate by centrifugation at 12,000 g for 10 min at 4°C. An aliquot of the soluble material was removed for input DNA and stored at −20°C. To reduce the SDS concentration to 0.1%, the remaining soluble material was diluted 10-fold in ChIP dilution buffer (1.2 mM EDTA, 16.7 mM Tris-HCl at pH 8.0, 167 mM NaCl, 1.1% Triton X-100, 0.01% SDS, plus PhosSTOP phosphatase inhibitor and cOmplete protease inhibitor cocktail [Roche, Basil, Switzerland]). To reduce non-specific background, the diluted soluble material was pre-cleared with Dynabeads protein G beads (#10003D, ThermoFisher, Waltham, MA), and, at the same time, the antibodies were linked to Dynabeads protein G beads (#10003D, ThermoFisher) by rotating for one hour at 4°C. At this point, the pre-cleared diluted soluble material was incubated with the antibody-bead mixtures, rotating at 4°C overnight. Immunoprecipitated material was washed three times with Low Salt Wash Buffer (2 mM EDTA, 20 mM Tris-HCl at pH 8.0, 150 mM NaCl, 1% Triton X-100, 0.1% SDS), three times with High Salt Wash Buffer (2 mM EDTA, 20 mM Tris-HCl at pH 8.0, 500 mM NaCl, 1% Triton X-100, 0.1% SDS), three times with LiCl Wash Buffer (1 mM EDTA, 1 mM Tris-HCl at pH 8.0, 1% DOC, 1% NP-40, 250 mM LiCl), and three times with TE buffer (10 mM Tris·Cl, pH 8.0; 1 mM EDTA). DNA complexes were eluted 30 min at 65°C with TE-SDS (10 mM Tris·Cl, pH 8.0; 1 mM EDTA; 1% SDS) and decrosslinked overnight at 65°C, along with input DNA, with the addition of 125 mM NaCl. Decrosslinked DNA complexes and input DNA were treated with RNaseA, and subsequently with proteinase K. Finally, immunoprecipitated and input DNA were purified with phenol:chloroform:isoamyl extraction (25:24:1), recovered by precipitation with ethanol in the presence of 300 mM NaOAc pH 5.2 and 2 μl of glycogen carrier (10 mg/ml), and resuspended in UltraPure DNase/RNase-Free Distilled Water (ThermoFisher) for later use. Libraries of immunoprecipitated DNA and input DNA were prepared using the NEBNext ChIP-seq Library Prep Master Mix Set for Illumina (#E6240, New England Biolabs, Ipswich, MA) according to the manufacturer's protocol. The quality and profile of the libraries was analyzed using Agilent High Sensitivity DNA Kit (#5067–4626, Agilent, Santa Clara, CA) and quantified using KAPA Library Quantification Kit (#KK4824, Kapa Biosystems, Wilmington, MA). Deep sequencing (100 bp paired-end) of the adult libraries – two biological replicates for H3K4me3, H3K4me1, H3K36me3, H3K27me3, RNAPII, input DNA and no biological replicates for H3K27ac and total histone H3 – was performed by the Macrogen Oceania NGS Unit on Illumina HiSeq 2000 instrument (Illumina, San Diego, CA, United States). Deep sequencing (40 bp paired-end) of the larva libraries – no biological replicates for H3K4me3, H3K4me1, H3K27me3, H3K27ac, RNAPII, input DNA – was performed by the Central Analytical Research facility (CARF), Brisbane, Queensland, Australia, on Illumina NextSeq 500 instrument (Illumina, San Diego, CA, United States).

## ChIP-seq data analyses

Adult raw Illumina sequencing reads were checked using FastQC v0.52 (http://www.bioinformatics. babraham.ac.uk/projects/fastqc/) and quality filtered using Trimmomatic v1.0.0 (SLIDINGWINDOW: 4:15, LEADING: 3, TRAILING: 3, HEADCROP: 5, MINLEN: 50) (RRID:SCR_011848) (*Bolger et al., 2014*). Quality filtered paired-end Illumina sequencing reads were then aligned to the *Amphimedon* genome (*Srivastava et al., 2010*) using Bowtie v1.1.2 (RRID:SCR_005476) (*Langmead et al., 2009*) with -m 1, -n 2, -X 500, –best parameters (uniquely mapped reads and maximum of two mismatches within the seed). Non-aligned reads were removed using SAMtools v0.1.19 (RRID:SCR_002105) (*Li et al., 2009*). For all the ChIP-seq data sets, strand cross-correlation measures were used to estimate signal-to-noise ratios using SPP v1.11.0 (RRID:SCR_001790). ChIP-seq data sets for each mark were flagged if the scores were below a normalized strand cross-correlation coefficient (NSC) threshold of 1.05, as described in the modENCODE and ENCODE guidelines (*ENCODE Project Consortium, 2012*; *Landt et al., 2012*; *Kellis et al., 2014*; *Kharchenko et al., 2008*). These analyses were performed on Galaxy-qld server (http://galaxy-qld.genome.edu.au/galaxy) developed within the GVL project (*Afgan et al., 2015*, *2016*) and maintained by the Research Computing Centre, University of Queensland, Australia.

Pearson's correlation coefficients (Pearson's r) of genome-wide fold enrichment (FE) signals (see below) was computed for biological replicates and a minimum threshold of 0.5 was required, as per *Ho et al. (2014)*. In addition, to ensure consistency between biological replicates, we further required an Irreproducible Discovery Rate (IDR) of at least 0.5 (see below), as described in the mod-ENCODE and ENCODE guidelines (*ENCODE Project Consortium, 2012*; *Landt et al., 2012*; *Kellis et al., 2014*; *Kharchenko et al., 2008*). ChIP-seq data sets that met these criteria were then merged across biological replicates (see ,).

Histone PTM regions of enrichment relative to corresponding sequenced input DNA controls were determined using MACS2 v2.1.0 (RRID:SCR_013291) (*Zhang et al., 2008*) according to mod-ENCODE, ENCODE and Roadmap Epigenomics consortiums guidelines (*Kundaje et al., 2015*; *ENCODE Project Consortium, 2012*; *Landt et al., 2012*; *Kellis et al., 2014*; *Kharchenko et al., 2008*). Specifically, MACS2 was used in broadpeak mode with a broadpeak *P*-value threshold of 0.1 and a narrowpeak threshold of 0.01 (-p 0.01, –broad, –nomodel, –extsize 146, -g 1.45e8). Enriched regions were scored on individual replicates (R1 and R2), pooled data (reads pooled across biological replicates) (P) and on subsampled pseudoreplicates (obtained by pooling reads from biological replicates and randomly subsampling, without replacement, two pseudoreplicates with half the total number of pooled reads) (PR1 and PR2). For each histone PTM, we defined 'R' as the set of peaks in P that overlap peaks in R1 and R2, and 'PR' as the set of peaks in P that overlap peaks in PR1 and PR2. Next, we defined 'M' as the set of peaks that match exactly in R and PR, and 'T' as the set of peaks that match exactly in R and PR as well as those that are unique to R or unique to PR. For a statement about reproducibility we required the M-to-T ratio to be at least 0.5 (*Figure 1—source data 4*). To obtain reliable regions of enrichment, we restricted all further analyses to enriched regions identified using pooled data that were also independently identified in both replicates and pseudoreplicates (the 'M' set). These regions of enrichment can be interpreted as high confidence regions, representing reproducible events across true biological replicates. For H3K27ac, for which no replication was available, we used the *P*-value column to rank peaks and only retained peaks with a p-value<0.001. We used the gappedPeak representation for the histone PTMs with relatively compact enrichment patterns, including H3K4me3, H3K27ac and H3K4me1. The gapped peaks are broad domains (passing *P*-value 0.1) that contain at least one narrow peak passing a *P*-value of 0.01. For the diffused histone PTMs – H3K36me3 and H3K27me3 – we used the broad-Peak representation. RNAPII peaks were detected using the peakzilla software (RRID:SCR_007471) (*Bardet et al., 2013*), using input DNA reads as control (-c 1.5, –s 3). The fraction of reads falling within peak regions (FRiP) was also calculated (see *Figure 1—source data 4*). In line with ENCODE guidelines (*ENCODE Project Consortium, 2012*; *Landt et al., 2012*; *Kellis et al., 2014*; *Kharchenko et al., 2008*), all our data sets have a FRiP enrichment of 1% or more.

For every pair of aligned ChIP and matching input DNA data sets, we also used MACS2 (*Zhang et al., 2008*) to generate genome-wide signal coverage tracks for every position in the *Amphimedon* genome (*Srivastava et al., 2010*). Input DNA was used as a control for signal normalization for the histone ChIP-seq coverage. The three types of signal score statistics computed per

base are as follows: (i) fold-enrichment ratio of ChIP-seq counts relative to expected background counts $_{local}$ (FE); (ii) negative log10 of the Poisson $P$-value of ChIP-seq counts relative to expected background counts $_{local}$ (ppois); and (iii) subtraction of noise from treatment sample (subtract).

Larva ChIP-seq data sets were analysed as described above, with the following minor modifications. Adapter contamination prior to read quality filtering was removed using Cutadapt (RRID:SCR_011841) (*Martin, 2011*). Reads were then quality filtered using Trimmomatic v1.0.0 (SLIDING-WINDOW: 4:15, LEADING: 3, TRAILING: 3, HEADCROP: 3, MINLEN: 20) (RRID:SCR_011848) (*Bolger et al., 2014*). Histone PTM and RNAPII regions of enrichment relative to sequenced input DNA controls were determined using MACS2 v2.1.0 (RRID:SCR_013291) (*Zhang et al., 2008*) in broadpeak mode with a broadpeak $q$-value threshold of 0.1 and a narrowpeak threshold of 0.05 (-q 0.05, –broad, –nomodel, –extsize 146, -g 1.45e8).

In both stages, chromatin states across the genome were defined using ChromHMM v1.10 (*Ernst and Kellis, 2012*), which is based on a multivariate Hidden Markov Model, using default parameters. For each ChIP-seq data set, read counts were computed in non-overlapping 200 bp bins across the *Amphimedon* genome (*Srivastava et al., 2010*). Each bin was discretised into two levels, one indicating enrichment and 0 indicating no enrichment. The binarization was performed by comparing ChIP-seq read counts to corresponding input DNA control read counts within each bin and using a Poisson $P$-value threshold of 1e-4 (the default discretization threshold in ChromHMM). We trained several models in parallel mode with the number of states ranging from 5 states to 15 states and chose a 9-state model as the best model that captures all the key interactions between the chromatin marks and cover all possible genomic locations (promoter, enhancer, gene body) that we expected to resolve given the selection of histone PTMs we used (H3K4me3, H3K27ac, H3K36me3, H3K4me1, H3K27me3 in adult; and H3K4me3, H3K27ac, H3K4me1, H3K27me3 in larva). To assign biologically meaningful mnemonics to the nine states, ChromHMM was used to compute the overlap and neighborhood enrichments of each state relative to various types of functional annotations (*Figure 1B*; *Figure 1—figure supplement 2*; *Figure 1—figure supplement 3*; *Figure 1—figure supplement 4*). State enrichment in different genomic features was calculated dividing the percentage of nucleotides occupied by a state in a particular genomic feature by the percentage of nucleotides that this genomic feature represents in the entire genome. For the overlap enrichment plots in the figures, the enrichments for each genomic feature (column) across all states is normalized by subtracting the minimum value from the column and then dividing by the max of the column. So, the values always range from 0 (white) to 1 (dark blue) (*i.e.*, a column wise relative scale). For the neighborhood positional enrichment plots, the normalization is done across all columns (*i.e.*, the minimum value over the entire matrix is subtracted from each value and divided by the maximum over the entire matrix). The functional annotations used were as follows: (1) CpG islands obtained using Hidden Markov Models as described in *Wu et al. (2010)*. (2) Exons, genes, introns, transcription start sites (TSSs) and transcription end sites (TESs), 200 bp windows around TSSs and 200 bp windows around TESs based on Aqu2.1 gene model annotations (*Fernandez-Valverde et al., 2015*). (3) Expressed and repressed genes, their TSSs and TESs. Genes were classified into expressed (CEL-seq normalized counts > 0.5) and repressed (CEL-seq normalized counts < 0.5) class based on their CEL-seq expression levels in the relevant stage (larva or adult) (*Levin et al., 2016*; *Hashimshony et al., 2012*; *Anavy et al., 2014*).

Regions of enrichment of the various histone H3 PTMs and RNAPII were overlapped with protein-coding genes and the Bioconductor R package GeneOverlap v1.14.0 (https://www.bioconductor.org/packages/release/bioc/html/GeneOverlap.html) was used to test and visualise their association with lists of various gene expression groups (*R Core Team, 2014*) (*Figure 2B*; *Figure 2—figure supplement 2B*). Protein-coding genes were classified into 'high', 'mid', 'low' and 'non-expressed' based on their CEL-seq expression levels in the relevant stage (larva or adult) (*Levin et al., 2016*; *Hashimshony et al., 2012*; *Anavy et al., 2014*). Expressed genes were liberally defined as genes that had CEL-seq read counts > 0 in the relevant stage. Specifically, to define 'high', 'medium', 'low' expressed genes, all protein-coding genes expressed in the relevant stage were sorted based on CEL-seq data values and separated into three bins of an equal number of genes, similar to previous analyses (*Schwaiger et al., 2014*).

Enhancer elements were predicted as reliable H3K4me1 regions of enrichment, which did not overlap TSSs (no intersection with 200 bp upstream or 200 bp downstream of the TSSs of protein-coding genes and lncRNAs), but overlapped with regions designated as being in an enhancer

chromatin state ('TxEnhA' or 'EnhWk' or 'EnhP' state in adult; 'TxEnhA1' or 'TxEnhA2' or 'EnhWk' or 'EnhP' state in larva) based on the ChromHMM analysis. The activated enhancer elements were predicted intersecting enhancer elements with H3K27ac significant peaks, requiring a 50% minimal overlap fraction. BEDTools v2.23.0 (RRID:SCR_006646) (*Quinlan and Hall, 2010*) was used to calculate overlaps between regions of enrichment and chromatin states with the different genomic features, as well as to identify the nearest TSS for each of the activated enhancer elements.

*De novo* motif enrichment analyses were performed using MEME-ChIP against JASPAR CORE and UniPROBE Mouse databases (-meme-minw 6, -meme-maxw 15, meme-nmotifs 20, -dreme-e 0.05, -meme-mod zoops) (RRID:SCR_001783) (*Machanick and Bailey, 2011*). Each motif was renamed according to their most similar motif in the TOMTOM database or literature, if any.

Gene Ontology (GO) functional enrichment analyses were performed using the Cytoscape plugin BiNGO (RRID:SCR_005736) (*Maere et al., 2005*; *Shannon et al., 2003*) with custom annotation and a FDR adjusted *P*-value cut-off of 0.01. All *Amphimedon* predicted peptides (*Fernandez-Valverde et al., 2015*) were annotated using BLASTp (RRID:SCR_001010) (*Altschul et al., 1990*) (*E*-value of 0.001) against the non-redundant (nr) NCBI protein database. All proteins were also searched for protein motifs and signal peptides using InterProScan 5 (*Jones et al., 2014*) with default parameters. KEGG pathway annotations were obtained on the webserver BlastKOALA for the taxonomic group 'Animals' against the 'family_eukaryotes + genus_prokaryotes' database file, using default settings. Pathway analyses were performed with the BlastKOALA annotation files using the KEGG Mapper – Reconstruct pathway tool (*Kanehisa et al., 2016*).

Transcription Start Site (TSS) input DNA-normalised coverage profiles and heatmaps were calculated using ngs.plot v2.61 (RRID:SCR_011795) (*Shen et al., 2014*) and deepTools v2.4.1 (*Ramírez et al., 2016*). As above, protein-coding genes were classified into 'high', 'mid', 'low' and 'non-expressed' based on their CEL-seq expression levels in the relevant stage (larva or adult) (*Levin et al., 2016*; *Hashimshony et al., 2012*; *Anavy et al., 2014*). Expressed genes were liberally defined as genes that had CEL-seq read counts >0 in the relevant stage. Specifically, to define 'high', 'medium', 'low' expressed genes, protein-coding genes expressed in the relevant stage were sorted based on CEL-seq data values and separated into three bins of an equal number of genes, similar to previous analyses (*Schwaiger et al., 2014*).

Only lincRNAs found in scaffolds larger than 10 kb were used for all the analyses and, given the compact genome of *Amphimedon* (*Fernandez-Valverde and Degnan, 2016*), all the TSS analyses were restricted to non-overlapping protein-coding genes with an intergenic distance >1 kb that were found in scaffolds larger than 10 kb.

All genome browser figures were generated using a local instance of the UCSC genome browser (RRID:SCR_005780) (*Kuhn et al., 2013*).

## ChIP-quantitative PCRs (ChIP-qPCRs)

ChIP-quantitative PCRs (ChIP-qPCRs) were performed using the LightCycler 480 platform (Roche, Basil, Switzerland). ChIP (H3K4me1, H3K27ac, H3K4me3, H3K27me3) and Input DNA libraries were diluted in water, combined with LightCycler 480 SYBR green I master mix (Roche, Basil, Switzerland) and 0.2 µM primers, then cycled with the following profile: 95°C for 10 min, 40 cycles of 95°C for 10 s, 60°C for 10 s, 72°C for 20 s. Primer sequences are available in *Supplementary file 1*.

Quantification cycle (Cq) values were extrapolated from manufacturers software (version 1.5.1.6.1 SP2) using High Confidence settings. A melt curve and no template controls (ntc) were also run to ensure single amplicons were responsible for the fluorescent signal. The numerical value 3.32 ($log_2 10$, representing 10% of input chromatin) was subtracted from the Cq value of the input sample to generate the adjusted input Cq. Two different intergenic regions not bound by our histone PTMs of interest were used as negative controls. Double delta (dd) Cq analysis was computed (see *Figure 1—source data 5*).

Specifically, the following formulas were used to calculate fold increase in signal over background:

dCq_IP = Cq_IP - Cq_Intergenic
dCq_Input = Cq_Input - Cq_intergenic
ddCq = dCq_IP - dCq_Input
Fold Change = $2^{\wedge}(\text{-ddCq})$

## High- and low-variance genes in *Amphimedon*

CEL-seq raw reads were processed and mapped back to the *Amphimedon* genome using Bowtie (RRID:SCR_005476) (*Langmead et al., 2009*). We then compressed the 82 *Amphimedon* developmental samples, from early cleavage to adult, into 17 stages averaging the biological replicates for each developmental stage across them. Larval stages have been combined in two different groups (Larvae 0–7 hr and Larvae 6–50 hr), as these developmental time points only have one replicate per time point. To reduce noise, the protein-coding genes and long non-coding RNAs with an overall expression of less than 100 CEL-seq raw counts throughout the whole developmental time course were discarded. The CEL-seq raw gene counts were then normalized using variance stabilizing transformation in DEseq2 1.6.3 (RRID:SCR_000154) (*Love et al., 2014*) and the 15,000 most variable genes (14,698 protein coding genes + 301 lncRNAs) were extracted using median absolute deviation. The 14,698 protein-coding genes were then filtered to retain only non-overlapping protein-coding genes with detectable expression at adult stage (CEL-seq normalized counts > 0) with an intergenic distance >1 kb that were found in scaffolds larger than 10 kb. This resulted in a total number of 3,200 'high-variance' genes. The remaining expressed (CEL-seq normalized counts > 0 in adult) non-overlapping protein-coding genes with an intergenic distance >1 kb that were found in scaffolds larger than 10 kb were considered 'low-variance' genes (n = 3,999). To define low, medium and high, the 3,200 high-variance genes and 3,999 low-variance genes were sorted based on CELseq data values and separated into three bins of an equal number of genes.

## Regulated and stable genes in *Amphimedon*

CEL-seq raw reads were processed and mapped back to the *Amphimedon* genome using Bowtie (RRID:SCR_005476) (*Langmead et al., 2009*). Read counts were normalized by dividing by the total number of counted reads and multiplying by $10^6$. We then compressed the 82 *Amphimedon* developmental samples, from early cleavage to adult, into 17 stages averaging the biological replicates for each developmental stage across them. Larval stages have been combined in two different groups (Larvae 0–7 hr and Larvae 6–50 hr), as these developmental time points only have one replicate per time point. To reduce noise, only the protein-coding genes with an expression of at least four CEL-seq normalised counts in at least two developmental time points were retained. To define the transcriptional stability of protein-coding genes, the coefficient of variation of gene expression was calculated for each protein-coding gene (n = 15,146), as reported by *Pérez-Lluch et al. (2015a)*. For the TSS input DNA-normalised coverage plots, these 15,146 protein-coding genes were then filtered to retain only expressed (CEL-seq normalized counts > 0 in the relevant stage [larva or adult]) non-overlapping protein-coding genes with an intergenic distance >1 kb that were found in scaffolds larger than 10 kb. Finally, from the full ranking of these expressed protein-coding genes, we defined the bottom 1,000 genes with the lowest variation in expression during development as 'stable' genes and the top 1,000 genes with the highest variation in expression as strongly developmentally 'regulated' genes.

## Regulated and stable genes in *Nematostella vectensis*

Available ChIP-seq data sets on adult female polyps for H3K4me3 and corresponding input DNA controls were used (*Schwaiger et al., 2014*). Aligned ChIP and matching input DNA data sets and developmentally stable and regulated genes were generated using the same procedures as in the sponge (see above). To obtain gene and transcript quantifications, we mapped available RNA-seq data sets (*Helm et al., 2013*) to NveGenes2.0 gene models (http://www.cnidariangenomes.org/) using kallisto (*Bray et al., 2016*).

## Orthologs identification and phylogeny

Orthologs of *Drosophila* PcG components and associated factors were identified using BLASTp (RRID:SCR_001010) (*Altschul et al., 1990*) searches against the predicted proteomes of the selected species (*Figure 4—source data 1*) with a threshold *E*-value of 0.001 and taking a maximum of 5 hits per species. All the obtained protein hits were aligned using MAFFT with L-INS-i mode (RRID:SCR_011811) (*Katoh and Standley, 2013*). The alignments were automatically trimmed with trimAl v1.4 (151) in -automated1 mode. Resulting trimmed alignments were then used for phylogenetic inference using FastTree2 (*Price et al., 2010*) with -wag -cat 8 -gamma parameters. The phylogenetic

trees were inspected manually to discriminate which BLASTp hits formed monophyletic clades with the *Drosophila* query sequences. The same methodology was used to identify the conserved ancestral microsyntenic pairs taken from *Irimia et al. (2012)*, but using *Homo sapiens* sequences as query proteins. The phylogeny-validated *Amphimedon* ortholog pairs were manually checked for contiguity in the genome and those found in different scaffolds or with more than two intervening genes were removed from the subsequent analyses.

## Data access

*Amphimedon* ChIP-seq data sets have been deposited to the NCBI Gene Expression Omnibus (GEO) (RRID:SCR_007303) (*Edgar et al., 2002*) under accession number GSE79645. *Amphimedon* genome assembly ampQue1 was used throughout the study. CEL-seq data sets can be obtained from NCBI GEO (GSE54364) (*Anavy et al., 2014*). *Amphimedon* RNA-seq data sets can be downloaded at NCBI's SRA (RRID:SCR_004891) with accession SRP044247 (*Fernandez-Valverde et al., 2015*). *Nematostella vectensis* RNA-seq data sets can be downloaded at NCBI's SRA with accession SRP018739 (*Helm et al., 2013*). *N. vectensis* ChIP-seq data sets can be obtained from NCBI GEO (GSE46488) (*Schwaiger et al., 2014*). We used the following gene model data sets for all analyses. *A. queenslandica*: Aqu2.1 models (http://amphimedon.qcloud.qcif.edu.au/) (last accessed February 25, 2017) (*Fernandez-Valverde et al., 2015*), lncRNAs (http://amphimedon.qcloud.qcif.edu.au/lncRNAs/) (last accessed February 25, 2017) (*Gaiti et al., 2015*); *N. vectensis*: NveGenes2.0 models (http://www.cnidariangenomes.org/)(last accessed February 25, 2017).

## Acknowledgements

We thank Alex de Mendoza for constructive comments on the manuscript, orthologs identification and phylogenetic analyses, William Hatleberg for KEGG pathways analysis, Simon Blomberg for statistical consultation, Nicholas Rhodes and Igor Makunin for bioinformatics support, Emily Wong for critical reading of the manuscript, and Kevin Dudley for sequencing of the larva ChIP-seq libraries.

## Additional information

### Funding

| Funder | Grant reference number | Author |
| --- | --- | --- |
| Australian Research Council | FL110100044 | Bernard M Degnan |

The funders had no role in study design, data collection and interpretation, or the decision to submit the work for publication.

### Author contributions

FG, Conceptualization, Data curation, Formal analysis, Validation, Visualization, Methodology, Writing—original draft, Project administration, Writing—review and editing; KJ, Formal analysis, Validation, Methodology, Writing—review and editing; SLF-V, Conceptualization, Formal analysis, Supervision, Project administration, Writing—review and editing; KER, Validation, Methodology, Writing—review and editing, Performed ChIP-quantitative PCRs (ChIP-qPCRs). All authors agree with her inclusion and place in the author list; BMD, Conceptualization, Supervision, Funding acquisition, Writing—original draft, Project administration, Writing—review and editing; MT, Conceptualization, Formal analysis, Supervision, Methodology, Writing—original draft, Project administration, Writing—review and editing

### Author ORCIDs

Federico Gaiti, http://orcid.org/0000-0001-5111-8816
Miloš Tanurdžić, http://orcid.org/0000-0002-7564-0868

## Additional files

### Supplementary files
• Supplementary File 1. Primer sequences.

### Major datasets
The following dataset was generated:

| Author(s) | Year | Dataset title | Dataset URL | Database, license, and accessibility information |
|---|---|---|---|---|
| Gaiti F, Jindrich K, Fernandez-Valverde SL, Roper KE, Degnan BM, Tanurdzic M | 2017 | Landscape of histone modifications in a sponge reveals the origin of animal cis-regulatory complexity | https://www.ncbi.nlm.nih.gov/geo/query/acc.cgi?acc=GSE79645 | Publicly available at the NCBI Gene Expression Omnibus (accession no: GSE79645) |

The following previously published datasets were used:

| Author(s) | Year | Dataset title | Dataset URL | Database, license, and accessibility information |
|---|---|---|---|---|
| Anavy L, Levin M, Khair S, Nakanishi N, Fernandez-Valverde SL, Degnan BM, Yanai I | 2014 | A high-resolution Amphimedon queenslandica transriptomic timecourse | https://www.ncbi.nlm.nih.gov/geo/query/acc.cgi?acc=GSE54364 | Publicly available at NCBI Gene Expression Omnibus (accession no: GSE54364) |
| Fernandez-Valverde SL, Calcino AD, Degnan BM | 2015 | Amphimedon queenslandica deep developmental transcriptomes | https://trace.ncbi.nlm.nih.gov/Traces/sra/sra.cgi?study=SRP044247 | Publicly available at NCBI Sequence Read Archive (accession no: SRP044247) |
| Helm RR, Siebert S, Tulin S, Smith J, Dunn CW | 2013 | Characterization of differential transcript abundance through time during Nematostella vectensis development | http://sra.dnanexus.com/studies/SRP018739/experiments | Publicly available at NCBI Sequence Read Archive (accession no: SRP018739) |
| Schwaiger M, Schönauer A, Rendeiro AF, Pribitzer C, Schauer A, Gilles A, Schinko J, Renfer E, Fredman D, Technau U | 2013 | Evolutionary conservation of the eumetazoan gene regulatory landscape | https://www.ncbi.nlm.nih.gov/geo/query/acc.cgi?acc=GSE46488 | Publicly available at NCBI Gene Expression Omnibus (accession no: GSE46488) |

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
