## [Decision Letter]

Thank you for submitting your article "Landscape of histone modifications in a sponge reveals the origin of animal *cis*-regulatory complexity" for consideration by *eLife*. Your article has been reviewed by three peer reviewers, and the evaluation has been overseen by Robb Krumlauf as the Senior Editor and Reviewing Editor. The following individuals involved in review of your submission have agreed to reveal their identity: Veronica Hinman (Reviewer #1); Ulrich Technau (Reviewer #3).

The reviewers have discussed the reviews with one another and the Reviewing Editor has drafted this decision to help you prepare a revised submission.

The consensus opinion of the reviewers is that this study is of great potential interest and could make a significant contribution to the field. These analyses could bridge the gap between published studies on *Nematostella* and on *Capsaspora* because in the latter case there was no evidence for enhancer/promoter landscapes. Therefore, the study deals with an important question, is well-designed, and has potential for both high relevant study and impact. However, there are two major weaknesses that dampen enthusiasm: a) an important part of the paper is the epigenetic analysis but the quality of H3K4me1 data is poor and b) and the marks of the lincRNAs is weak. These issues must be addressed in a revision before the paper can be considered further. We normally like to see revisions made within two months but we appreciate that the issues raised may require a longer time period and we are willing to allow you the additional time if needed. This may mean that the resubmission is given a new manuscript number but we will make every effort to ensure that it is handled by the same initial reviewers.

A) Epigenetics: The H3K4me1 and H3K27ac data is of poor quality and there are a variety of issues with the technical aspects of the paper. This includes missing ChIP-qPCR validations, questionable antibody quality and an absence of supplementary figures dealing with data integrity.

Some of the reviewers feel it would be important to perform additional replicates of H3K4me1 and H3K27ac ChIPs. If this is possible it could enhance the paper and address the weakness of the existing data. However, we are willing to stop short of this request because of the time involved and instead request that more analysis and data be provided to confirm existing peaks. Minimally, you need to provide more thorough statistical analysis and test the effects of different thresholds on the chromatin states. This will not eliminate the issues associated with data noisiness (only new ChIPs can do that) but will allow the reader to gain a better insight into the quality of the data that was used to reach the conclusions. This involves:

1) Extensive qPCR (20 – 25 targets / ChIP) validations (including a number of negative controls).

2) Heatmaps (not average profiles) followed by HCL or K-means clustering of their ChIP-seq data over various genomic features (TSS, genes, newly identified chromHMM features), including both replicates side by side.

3) Thorough comparisons of the ChIP-seq data (correlation of signal in genomic bins for all the datasets and all replicates).

4) BAM files and signal files (BigWig) should be uploaded to GEO and made available for reviewers to independently assess data quality.

5) For all the relevant genomic examples (named above), both experimental replicates should be shown.

6) This paper, and all conclusions, rely entirely on the use of off-the-shelf antibodies. All of the data analyses derive from experiments with these antibodies. Therefore, I think these need to be "gold standard experiments." It’s reassuring that the proteins are 100% identical in the region that the antibody recognizes, but this does not indicate that these antibodies are specific. A great control would be to perform ChiP-PCR or western blot on control versus some stage treated with a histone methylation/acetylation inhibitors. This would confirm specificity.

7) Data for Figure 3. There needs to be some statistical analysis for the statement for the significance of the reduced H3K4me3 in dynamically expressed vs. stable. I don't agree with the finding for the sponge is similar to *Nematostella* (Figure 3AvsC). The difference between the top bin and the bottom bin is striking in *Nematostella* and much more modest, perhaps even not significant, in the sponge. This however is where a genuine statistical analysis is needed to make the stronger point. If this is a qualitative analysis the authors should be clear about this in the discussion of this data. I also wonder here is this analysis is confounded by inclusion of the many genes for which the one peak of expression is in the adult (Figure 3-1) These could be genes that are nonetheless very stably expressed in the adult, but the resolution of the time-course assigns "adult" as one time equivalent to very restricted developmental stages. This analysis might be better done by considering only the developmental stages, and might provide a better significance.

B) lincRNAs: There is a concern about the ability to reach conclusions the lincRNA aspect of the study based on figures presented. This could simply be omitted.

---

## [Author Response]

*The consensus opinion of the reviewers is that this study is of great potential interest and could make a significant contribution to the field. These analyses could bridge the gap between published studies on Nematostella and on Capsaspora because in the latter case there was no evidence for enhancer/promoter landscapes. Therefore, the study deals with an important question, is well-designed, and has potential for both high relevant study and impact. However, there are two major weaknesses that dampen enthusiasm: a) an important part of the paper is the epigenetic analysis but the quality of H3K4me1 data is poor and b) and the marks of the lincRNAs is weak. These issues must be addressed in a revision before the paper can be considered further. We normally like to see revisions made within two months but we appreciate that the issues raised may require a longer time period and we are willing to allow you the additional time if needed. This may mean that the resubmission is given a new manuscript number but we will make every effort to ensure that it is handled by the same initial reviewers.*

*A) Epigenetics: The H3K4me1 and H3K27ac data is of poor quality and there are a variety of issues with the technical aspects of the paper. This includes missing ChIP-qPCR validations, questionable antibody quality and an absence of supplementary figures dealing with data integrity.*

*Some of the reviewers feel it would be important to perform additional replicates of H3K4me1 and H3K27ac ChIPs. If this is possible it could enhance the paper and address the weakness of the existing data. However, we are willing to stop short of this request because of the time involved and instead request that more analysis and data be provided to confirm existing peaks. Minimally, you need to provide more thorough statistical analysis and test the effects of different thresholds on the chromatin states. This will not eliminate the issues associated with data noisiness (only new ChIPs can do that) but will allow the reader to gain a better insight into the quality of the data that was used to reach the conclusions.*

Thank you for these comments and recommendations. We also agree that additional ChIPs would not be necessary, especially given the larval ChIPs corroborate the adult ChIPs despite being a different stage comprised of different cell types with markedly overall different transcriptional profiles. We agree that without this confirmation additional replicates would be necessary. In the revision, we have made sure that we highlight, wherever possible, the consistencies between larval and adult ChIP results. Nonetheless, as outlined below, we have undertaken a range of new validations and we have extended our statistical analyses, and included new figure plates and statements.

*This involves:*

*1) Extensive qPCR (20 – 25 targets / ChIP) validations (including a number of negative controls).*

As per request of the reviewers, we have now performed ChIP-qPCRs validations of our ChIP-seq data sets for H3K27ac (n =20 targets) and H3K4me1 (n =21 targets) to confirm existing peaks. In addition, we have validated a subset of H3K4me3 (n =10 targets) and H3K27me3 (n =10 targets) peaks. For each of these four histone PTMs, ChIP-qPCRs confirmed our ChIP-seq findings for >80% of the targets, compared to negative controls (two different intergenic regions not bound by our histone PTMs of interest). Importantly, in the great majority of the cases, we observed high signal when corrected for background.

Specifically, in the case of H3K4me3, we designed primers to target regions of enrichment at transcription start sites (TSSs) of genes identified using our peak calling strategy. We were able to validate 90% of the targets. For H3K27me3, we designed primers to target transcriptionally silenced regions of enrichment and validated 80% of them. Most importantly, for H3K4me1 and H3K27ac we designed primers to predominantly target regions of enrichment corresponding to our *in-silico* predicted enhancer elements, and were able to successfully validate >85% and >95% of them, respectively.

At the end of this response letter, we provide the methods on how these experiments were performed. We also have added to this revision, as a ‘Related Manuscript File’, a file that shows the ChIP-qPCR results. We have elected not to include these data in the paper as such confirmation experiments for histone H3 modifications are not typically found in equivalent reports from basal metazoans (i.e., *Nematostella*) or unicellular holozoans (i.e., *Capsaspora*). If requested, we can add these data to the manuscript.

*2) Heatmaps (not average profiles) followed by HCL or K-means clustering of their ChIP-seq data over various genomic features (TSS, genes, newly identified chromHMM features), including both replicates side by side.*

In addition to Figure 2—figure supplement 1, where we have shown the TSS-centred average input DNA-normalised read coverage heatmaps of RNAPII, H3K27ac, H3K36me3, H3K4me1 and H3K27me3 across *Amphimedon*protein-coding genes, and heatmaps in Figure 5, showing the input DNA-normalised coverage of H3K4me1, H3K4me3 and their ratio in adult *Amphimedon*across TSSs of lincRNAs, we have now added heatmaps over the newly identified activated enhancer elements in Figure 6. This demonstrates the higher enrichment of H3K4me1 and H3K27ac over H3K4me3 in active enhancers, a typical biochemical signature of eumetazoan enhancers (Schwaiger et al. 2014), and also represents an independent confirmation of H3K4me1 signal over background.

*3) Thorough comparisons of the ChIP-seq data (correlation of signal in genomic bins for all the datasets and all replicates).*

We re-evaluated reproducibility of high-throughput experiments by measuring consistency between the biological replicates within our ChIP-seq experiments. First, Pearson’s correlation coefficients (Pearson’s r) of genome-wide fold enrichment (FE) signals was computed for biological replicates and a minimum threshold of 0.5 was required. In addition, to ensure consistency between biological replicates, we further required an Irreproducible Discovery Rate (IDR) of at least 0.5, as described in the modENCODE and ENCODE guidelines (Landt et al. 2012). ChIP-seq data sets that met these criteria were then merged across biological replicates. This is now stated in the Materials and methods section:

“Pearson’s correlation coefficients (Pearson’s r) of genome-wide fold enrichment (FE) signals (see below) was computed for biological replicates and a minimum threshold of 0.5 was required, as per Ho et al. (Ho et al., 2014). In addition, to ensure consistency between biological replicates, we further required an Irreproducible Discovery Rate (IDR) of at least 0.5 (see below), as described in the modENCODE and ENCODE guidelines (19, 127-129). ChIP-seq data sets that met these criteria were then merged across biological replicates (see [Supplementary-material SD4-data]).”

A complete summary statistics and quality metrics of the ChIP-seq datasets used in this study is shown in [Supplementary-material SD4-data].

In addition, we have now incorporated a supplementary figure addressing data integrity. Specifically, in Figure 1—figure supplement 2 we have now shown Pearson correlation coefficients between adult histone modifications and RNA Polymerase II experiments and input DNA-normalized coverage of each biological replicate (R1 and R2) for each of the different histone modifications side by side. Note that H3K36me3 was flagged for low signal to noise, potentially explaining the somewhat high correlation with H3K27me3, and RNAPII replicate 1 did not pass the quality threshold required so it has been excluded from all further analyses. However, this does not affect the conclusions of the paper in any way. Lastly, as explained in detail in comment #1 above, we have performed ChIP-qPCRs validations of our ChIP-seq data sets for H3K4me3, H3K27ac, H3K4me1 and H3K27me3.

Together, this evidence provides further support for the quality of the data that we used to reach the main conclusions of our study.

*4) BAM files and signal files (BigWig) should be uploaded to GEO and made available for reviewers to independently assess data quality.*

GEO GSE79645 has now been updated as requested. Please note that GEO does not accept alignment files (BAM) as processed data, as indicated at http://www.ncbi.nlm.nih.gov/geo/info/seq.html [the further processed data (e.g., wig/peak files) should be supplied instead]. As per GEO guidelines, all raw data files (fastq) along with peak files (BED), and signal files (wig) are now available at the following link, allowing review of record GSE79645 while it remains in private status: https://www.ncbi.nlm.nih.gov/geo/query/acc.cgi?token=qxyhmmiajrolnqt&acc=GSE79645 (Last accessed February 25, 2017).

*5) For all the relevant genomic examples (named above), both experimental replicates should be shown.*

As mentioned above in comment #3, we have now incorporated these into a supplementary figure addressing data integrity. In Figure 1—figure supplement 2 we have now shown input DNA-normalized coverage of each biological replicate (R1 and R2) for each of the different histone modifications side by side for five different genomic regions, either transcriptionally active or silenced. Apart from RNAPII replicate 1, which did not pass the quality threshold required so it has been excluded from all further analyses, we obtained highly reproducible data sets. In addition, in all relevant genomic examples presented in the figures, only the high confidence regions of enrichment representing reproducible events across true biological replicates are shown. For these reasons, we do not think it is necessary to show both experimental replicates in each and every one of the genomic examples.

*6) This paper, and all conclusions, rely entirely on the use of off-the-shelf antibodies. All of the data analyses derive from experiments with these antibodies. Therefore, I think these need to be "gold standard experiments." It’s reassuring that the proteins are 100% identical in the region that the antibody recognizes, but this does not indicate that these antibodies are specific. A great control would be to perform ChiP-PCR or western blot on control versus some stage treated with a histone methylation/acetylation inhibitors. This would confirm specificity.*

We have followed current approaches adopted by previous studies in non-bilaterian model organisms (*e.g.*, Schwaiger et al. 2014; Sebé-Pedrós et al. 2016) where sequence conservation has been deemed sufficient to justify the usage of antibodies against the N-terminal tail of histone H3 in these organisms. In fact, the antibodies used in our study recognize the correct epitopes in even more distantly related eukaryotes, where their usage has been successfully reported [e.g., *Capsaspora*(Sebé-Pedrós et al. 2016), *Phaseolusvulgaris*(Barraza et al. 2015), yeast (Harmeyer et al. 2015), and *Tetrahymena*(Liu et al. 2007)]. Although we agree with the reviewers that the conservation of the entire amino acids sequence of histone H3, along with the relevant histone methyltransferase and acetyltransferase, which we also show, does not per seindicate that the antibodies used for chromatin immunoprecipitation are specific, we believe that the ‘gold standard’ had already been set by rich body of literature that had not attracted any major criticisms over the years.

*7) Data for Figure 3. There needs to be some statistical analysis for the statement for the significance of the reduced H3K4me3 in dynamically expressed vs. stable. I don't agree with the finding for the sponge is similar to Nematostella (Figure 3AvsC). The difference between the top bin and the bottom bin is striking in Nematostella and much more modest, perhaps even not significant, in the sponge. This however is where a genuine statistical analysis is needed to make the stronger point. If this is a qualitative analysis the authors should be clear about this in the discussion of this data. I also wonder here is this analysis is confounded by inclusion of the many genes for which the one peak of expression is in the adult (Figure 3-1) These could be genes that are nonetheless very stably expressed in the adult, but the resolution of the time-course assigns "adult" as one time equivalent to very restricted developmental stages. This analysis might be better done by considering only the developmental stages, and might provide a better significance.*

We agree with the reviewers that a genuine statistical analysis is needed to make the stronger point. We have now compared the log2(H3K4me3/input DNA) distributions between regulated and stable genes in both *Amphimedon*and *Nematostella*and performed a Mann-Whitney U test to show that the difference in H3K4me3 enrichment between stable and regulated genes is statistically significant. This is now mentioned in the main text. Specifically:

“Although stable and regulated genes had similar levels of RNAPII and total histone H3 (Figure 3—figure supplement 2A, 2B), the stable genes were strongly marked by H3K4me3 and the regulated genes had significantly lower levels of H3K4me3 (Mann-Whitney U test, P-value =7.431e-05; Figure 3), suggesting that reduction in H3K4me3 levels does not affect expression of the regulated genes (Perez-Lluch et al., 2015).”

And:

“Additionally, as shown in *Drosophila*(Perez-Lluch et al., 2015), regulated genes showed higher levels of H3K27me3 (Mann-Whitney U test, P-value <6.517e-06) and lower levels of H3K36me3 (Mann-Whitney U test, P-value <9.235e-08) than did stable genes (Figure 3—figure supplement 2C, 2D). Analyzing RNA-seq–based gene expression through the development of the cnidarian *Nematostellavectensis*(Helm et al., 2013) and previously published ChIP-seq data sets in *Nematostella*adult female polyps (Schwaiger et al., 2014), we obtained the same pattern (Mann-Whitney U test, P-value <2.2e-16; Figure 3).”

*B) lincRNAs: There is a concern about the ability to reach conclusions the lincRNA aspect of the study based on figures presented. This could simply be omitted.*

We have now expanded this analysis by performing additional statistical analysis and showed that the difference in H3K4me1-to-H3K4me3 signal is significantly different between the two predicted subpopulations of lincRNAs in *Amphimedon*. Specifically:

“Similarly to Ilott et al. (Ilott et al., 2014), we arbitrarily adopted a H3K4me1-to-H3K4me3 ratio of >1.2 and <0.8 to define elincRNAs and plincRNAs, respectively. […] Lastly, 43 (20%) lincRNAs could not be assigned to either group, that is, 0.8<H3K4me1-to- H3K4me3<1.2 (Figure 5; [Supplementary-material SD9-data]; Figure 5—figure supplement 1).”

These results strongly resemble those recently found in *Capsaspora*lincRNAs (Sebé-Pedrós et al. 2016), where similar subpopulations of lincRNAs have been identified based on their H3K4me1- to-H3K4me3 signatures, and we think represent interesting findings that are worth reporting in the manuscript.